# RECONCILING IN-CONTEXT AND IN-WEIGHT LEARNING: A DUAL-SPACE MODELING PERSPECTIVE

## ABSTRACT

In-context learning (ICL) is a valuable capability exhibited by Transformers pretrained on diverse sequence tasks. However, prior studies have observed that ICL often exhibits a conflict with the model's inherent in-weight learning (IWL) capability. In this work, we aim to reconcile ICL and IWL by disentangling the model's encoding spaces for context and input samples. To do so, we first propose a dual-space modeling framework, explicitly modeling a task representation space via the dual space of the sample representation space. Such a dual-space structure can be derived from the linear representation hypothesis and, as we theoretically prove, is conducive to ICL by representation learning. Furthermore, we show that the standard Transformer architecture with softmax self-attention is inherently limited in realizing this structure. Building on this insight, we introduce CoQE, a Transformer architecture with separate context-query encoding, to realize the disentanglement between context and sample representations. Through experiments on both regression and classification tasks, we demonstrate that CoQE not only achieves lower ICL error compared to the standard Transformers, but also successfully reconciles ICL and IWL under diverse data distributions.

## 1 INTRODUCTION

In recent years, large-scale models based on the Transformer architecture have demonstrated remarkable capabilities across language (Brown et al., 2020; Guo et al., 2025), vision (Achiam et al., 2023; Maaz et al., 2024), and robotics (Driess et al., 2023; Zitkovich et al., 2023). Among these capabilities, the in-context learning (ICL) ability has drawn increasing attention, as it offers a general paradigm for task generalization. ICL refers to the capability of a pretrained Transformer model to solve previously unseen tasks by using demonstration examples in the prompt—without updating its parameters. In contrast, in-weight learning (IWL) characterizes the conventional ability of a model to recall the memory stored in weights. An ideal model would seamlessly integrate both capabilities: relying on memory to handle training tasks, while adapting to new tasks through contextual cues.

However, recent studies suggest that there exists an inherent conflict between ICL and IWL (Park et al., 2025; Nguyen & Reddy, 2025). This leads to a notable performance degradation when the demonstration examples deviate from the training distribution (Chan et al., 2025), thereby limiting the generalization ability of ICL. How to eliminate this conflict is thus a valuable question. Singh et al. (2023; 2025) suggested that their conflict may stem from competition between the two interwined capabilities for shared model circuits during training. Since ICL can be viewed as a context-based inference strategy, whereas IWL relies on representations of individual samples, it implies that the root cause of ICL-IWL conflict lies in the entangled nature of how Transformers encode context and sample-level information.

In this work, we hypothesize that the conflict between ICL and IWL can be resolved by explicitly disentangling the encoding processes for context and sample. To this end, we propose a theoretical framework that introduces a separate encoding space for the context defined as the *task representation space*, in contrast with the standard *sample representation space*. Notably, under the widely accepted linear representation hypothesis (Mikolov et al., 2013; Nanda et al., 2023; Park et al., 2024), we show that the relationship between the sample representation space and the task representation space can be modeled via a *dual-space* formulation. Building on this framework, we prove the completeness of a sample representation space under sufficient training tasks, which could facilitate

task generalization by ICL. Moreover, we formalize the entangled nature of Transformers' encoding process-standard softmax attention does not support such a dual-space structure, highlighting a contrast with linear attention mechanisms commonly adopted in recent theoretical analysis.

Motivated by our analysis, we propose a straightforward yet effective architecture, CoQE. Unlike standard Transformers, CoQE employs separate pathways to encode context and query samples, aiming to learn the task representation space and sample representation space, respectively. The final model output is obtained by computing the inner product between elements from the two spaces according to the Riesz representation theorem. We conduct extensive experiments on both regression and few-shot classification tasks. Our results show that CoQE not only achieves lower ICL error than Transformers in both in-distribution and out-of-distribution scenarios, but also robustly reconciles ICL and IWL, yielding Pareto improvements for both capabilities under diverse data distributions.

## 2 PRELIMINARIES

**In-context learning setup.** The basic setup for analyzing ICL was first introduced by Garg et al. (2022) and has since been widely adopted (Yadlowsky et al., 2023; Pan et al., 2023). Consider a distribution $\mathcal{D}_{\mathcal{X}}$ over an input space $\mathcal{X} \subseteq \mathbb{R}^{d_x}$, and let $\mathcal{F}$ denote a class of functions over a distribution $\mathcal{D}_{\mathcal{F}}$. For each prompt, we first sample a task $f \sim \mathcal{D}_{\mathcal{F}}$, then draw a set of $n$ input-output pairs $\{(\mathbf{x}_i, \mathbf{y}_i)\}_{i=1}^n$, where $\mathbf{x}_i \stackrel{\text{i.i.d.}}{\sim} \mathcal{D}_{\mathcal{X}}$ and $\mathbf{y}_i = f(\mathbf{x}_i)$. These sample pairs serve as context. Then, we independently generate a query input $\mathbf{x}_q \sim \mathcal{D}_{\mathcal{X}}$. The final prompt is gathered as a sequence:

$$\mathcal{P} = \big(\mathbf{x}_1, \mathbf{y}_1, \ldots, \mathbf{x}_n, \mathbf{y}_n, \mathbf{x}_q\big).$$

The ICL capability of a pretrained model $\mathbb{M}_\theta$ refers to its accuracy to produce predictions $\hat{\mathbf{y}}_q = \mathbb{M}_\theta(\mathcal{P})$ for $\mathbf{y}_q = f(\mathbf{x}_q)$, without having explicit knowledge of the current task $f$ and without updating its parameters.

Chan et al. (2022) extend this setting by introducing few-shot image classification tasks. In this setup, $\mathbf{x}$ represents an encoded image, and $\mathcal{F}$, as a set of classifiers, maps $\mathcal{X}$ to a finite label set $\mathcal{Y}$. The ICL capability refers to the model's ability to correctly classify a query image $\mathbf{x}_q$ based on the image-label pairs provided in the context.

**Transformer model.** A standard single-head self-attention layer (Vaswani et al., 2017) operates on an input matrix $Z \in \mathbb{R}^{d_e \times L}$, where $L$ is the sequence length and $d_e$ the embedding dimension. Let $Q = W_Q Z$, $K = W_K Z$, $V = W_V Z$ with $W_Q, W_K \in \mathbb{R}^{d_k \times d_e}$ and $W_V \in \mathbb{R}^{d_v \times d_e}$. The attention output is

$$\mathrm{SA}(Z) = Z + W_O V \cdot \mathrm{softmax}\left(\frac{K^\top Q}{\sqrt{d_k}}\right),$$

where $W_O \in \mathbb{R}^{d_e \times d_v}$ and the softmax is applied column-wise. This operation can be applied to sequences of arbitrary length, and multi-head attention concatenates several such outputs before a linear projection.

For the theoretical analysis of ICL, the prompt $\mathcal{P}$ is typically re-organized into an embedding matrix:

$$Z = \begin{pmatrix} \mathbf{x}_1 & \cdots & \mathbf{x}_n & \mathbf{x}_q \\ \mathbf{y}_1 & \cdots & \mathbf{y}_n & 0 \end{pmatrix} \in \mathbb{R}^{(d_x+1) \times (n+1)},$$

where $d_x$ is the input feature dimension. Moreover, they often use a linear self-attention variant (LSA) obtained by removing the softmax and merging parameters:

$$\mathrm{LSA}(Z) = Z + \frac{1}{n} W_{OV} Z Z^\top W_{KQ} Z,$$

where $W_{OV} = W_O W_V$, $W_{KQ} = W_K^\top W_Q \in \mathbb{R}^{(d_x+1) \times (d_x+1)}$ are trainable, and $1/n$ is a scaling constant. The model prediction $\hat{\mathbf{y}}_q$ for the query is taken as the bottom-right entry of $\mathrm{LSA}(Z)$.

**Dual space.** Before formally introducing our dual-space modeling framework, we first present the general mathematical definition of the dual space.

**Definition 2.1** (Dual space). *Let $V$ be a finite-dimensional inner product space over a field $\mathbb{F}$ (typically $\mathbb{R}$ or $\mathbb{C}$) with inner product $\langle \cdot, \cdot \rangle$. The* dual space *of $V$, denoted $V^*$, is the set of all linear functionals from $V$ to $\mathbb{F}$:*

$$V^* \triangleq \{f : V \to \mathbb{F} \mid f \text{ is linear}\}. \tag{1}$$

For every $f \in V^*$, there exists a unique vector $\omega \in V$, called the Riesz representation of $f$, such that

$$f(v) = \langle \omega, v \rangle, \quad \forall v \in V.$$

Let $\{e_1, \ldots, e_n\}$ be the basis of $V$. The dual basis $\{e^1, \ldots, e^n\} \subset V^*$ is defined by

$$e^i(e_j) = \delta_{ij}, \quad 1 \leq i, j \leq n,$$

where $\delta_{ij}$ is the Kronecker delta.

In the following, we will show that this dual-space formulation can be used to model the relationship between a task representation space and the model's sample representation space. Moreover, by the Riesz representation theorem, elements from the two spaces can be composed via inner product.

## 3 DUAL-SPACE MODELING FRAMEWORK

In this section, we present our main theoretical results, including the dual-space modeling of the sample representation space and the task representation space, as well as the resulting implications for the model's representation learning and generalization error. Then we turn to the ICL setting and further discuss how LSA and SA behave differently under our proposed framework.

### 3.1 TASK REPRESENTATION SPACE

We begin with the widely acknowledged linear representation hypothesis (Mikolov et al., 2013; Park et al., 2024), from which we formalize the definition of a linear sample representation space.

**Definition 3.1** (Linear sample representation space). *Let $\mathcal{X} \subseteq \mathbb{R}^{d_x}$ denote the input space and $\mathcal{Y}$ the label set. A linear sample representation space $\mathcal{M} \subseteq \mathbb{R}^d$ is a finite-dimensional inner product space equipped with a mapping $\phi : \mathcal{X} \to \mathcal{M}$, such that*

1. *(Learnability) $\phi$ is parameterized by a model $\mathbb{M}$ and can be learned from data;*

2. *(Linear Measurement) In the case of regression with $\mathcal{Y} \subseteq \mathbb{R}$, there exists a linear transformation $\omega$ such that, given any $(\mathbf{x}, \mathbf{y})$ pair, the label can be expressed as*

$$\mathbf{y} = \langle \omega, \phi(\mathbf{x}) \rangle. \tag{2}$$

*In the case of classification with $\mathcal{Y} = \{0, 1\}$, the label probability is given by*

$$\mathrm{logit}\mathbb{P}(\mathbf{y} = 1 \mid \mathbf{x}) = \langle \omega, \phi(\mathbf{x}) \rangle. \tag{3}$$

Definition 3.1 formalizes the notion of a sample representation space under the linear representation hypothesis in the single-task setting. We then extend to the multi-task case, assuming that there exists a shared linear sample representation space across tasks. Note that this assumption has been implicitly embedded in a wide range of theoretical and algorithmic work (Caruana, 1997; Hu et al., 2023; Zhang et al., 2024b). Based on this assumption, we define the corresponding linear task transformation space. Without loss of generality, we consider only the regression case.

**Definition 3.2** (Linear task transformation space). *Let $\mathcal{F} = \{f : \mathcal{X} \to \mathbb{R}\}$ denote a task function space defined over the input space $\mathcal{X}$. We assume that there exists a sample representation space $\mathcal{M}_{\mathcal{F}} \subseteq \mathbb{R}^d$, together with a mapping $\phi_{\mathcal{F}}$, such that $\mathcal{M}_{\mathcal{F}}$ is linear with respect to $\mathcal{X}$ and each label set $\mathcal{Y}_f = \{f(\mathbf{x}) \mid \mathbf{x} \in \mathcal{X}\}$, $\forall f \in \mathcal{F}$. A linear task transformation space is then defined as a linear functional space $\mathcal{T} = \{t : \mathcal{M}_{\mathcal{F}} \to \mathbb{R}\}$, equipped with a mapping $\psi : \mathcal{F} \to \mathcal{T}$ such that for any $f \in \mathcal{F}, \psi(f) = t$ satisfying*

$$f(\mathbf{x}) = t(\phi_{\mathcal{F}}(\mathbf{x})), \quad \forall \mathbf{x} \in \mathcal{X}. \tag{4}$$

Building upon this foundation, we next introduce a novel perspective: to model the task transformation space as the dual space of the sample representation space.

**Proposition 3.3** (Task-sample duality). *Let $\mathcal{X}$ be the input space and $\mathcal{Y}_f$ the multiple label sets corresponding to each task $f \in \mathcal{F}$. Under Definition 3.2, there exists a linear sample representation space $\mathcal{M}_{\mathcal{F}}$ and a linear task transformation space $\mathcal{T}$, where $\mathcal{T}$ is the dual space of $\mathcal{M}_{\mathcal{F}}$, i.e. $\mathcal{T} = \mathcal{M}_{\mathcal{F}}^*$.*

**Definition 3.4** (Task representation space). *Under Proposition 3.3, for each task $f \in \mathcal{F}$, $\psi(f) \in \mathcal{T}$ admits a unique Riesz representation $\omega_f$. The task representation space $\mathcal{W}_{\mathcal{F}}$ is defined as the set of all such Riesz representations. Then for any $f \in \mathcal{F}$, we have*

$$f(\mathbf{x}) = \langle \omega_f, \phi_{\mathcal{F}}(\mathbf{x}) \rangle, \quad \forall \mathbf{x} \in \mathcal{X}. \tag{5}$$

In summary, we map various nonlinear tasks within a multi-task setting to vectors in the task representation space, leveraging the linear representation hypothesis, the dual-space formulation, and the Riesz representation theorem. From the above formulation, we can further define basis representations and basis transformations, along with the relationship between them.

**Definition 3.5** (Basis task representations). *Under Proposition 3.3, let $\{m_1, \ldots, m_d\}$ be a basis of the sample representation space $\mathcal{M}_{\mathcal{F}}$, and let $\{t_1, \ldots, t_d\}$ be the corresponding dual basis of the task transformation space $\mathcal{T}$. The basis task representations are defined as the Riesz representations of $\{t_1, \ldots, t_d\}$, denoted by $\{\omega_1, \ldots, \omega_d\}$, which satisfy*

$$\langle \omega_i, m_j \rangle = \delta_{ij}, \quad 1 \leq i, j \leq d. \tag{6}$$

Thus, every sample representation $\phi_{\mathcal{F}}(\mathbf{x})$ can decompose uniquely as $\phi_{\mathcal{F}}(\mathbf{x}) = \sum_{i=1}^{d} \alpha_i(\mathbf{x}) m_i$, and every task representation $\omega_f$ can decompose uniquely as $\omega_f = \sum_{j=1}^{d} \beta_j \omega_j$. The output can be given by the bilinear pairing

$$\langle \omega_f, \phi_{\mathcal{F}}(\mathbf{x}) \rangle = \sum_{i=1}^{d} \alpha_i(\mathbf{x}) \beta_i.$$

Our modeling provides a new insight: in representation learning, defining or identifying a basis for the sample representation space is a common practice, where each basis often corresponds to an independent attribute or concept (e.g., gender, identity). However, a natural question arises: why do certain attributes correspond to basis sample representations, while others do not? Beyond heuristic judgments about attribute importance, our modeling provides a principled explanation: basis sample representations and basis task representations are corresponding and mutually defining. In other words, if an attribute corresponds to a basis sample representation, then it must also correspond to solving a specific basis task.

Our next Theorem 3.6 shows that, under the dual-space modeling framework, a sufficient set of tasks guarantees a basis-covering sample representation space. We also provide a generalization error bound under our modeling framework in Theorem 3.7.

**Theorem 3.6** (Completeness of basis representations under task traversal). *Under Proposition 3.3, we assume that a learner with sample representation mapping $\phi_\theta$ is presented with a task traversal curriculum $\mathcal{C}$ such that: $\text{span}\{t \mid t \in \mathcal{C}\} = \mathcal{T}$. Then, if the learner achieves zero empirical error, the learned representation mapping $\phi_\theta$ satisfies: $\text{span}\{\phi_\theta(\mathbf{x}) \mid \mathbf{x} \in \mathcal{X}\} = \mathcal{M}_{\mathcal{F}}$; equivalently, each basis sample representation $m_i$ occurs in $\phi_\theta$.*

**Theorem 3.7** (Generalization error bound). *Under Proposition 3.3 and Definition 3.4, for any task $f$ represented by $\omega_f$ and input $\mathbf{x}$ represented by $\phi_{\mathcal{F}}(\mathbf{x})$, the predictor is $\hat{y} = \langle \omega_f, \phi_{\mathcal{F}}(\mathbf{x}) \rangle$. We assume that (1) $\|\omega_f\|_2 \leq 1, \forall f \in \mathcal{F}$; (2) the feature map is isotropic: for an orthonormal basis $\{m_j\}_{j=1}^{d}$ of $\mathcal{M}_{\mathcal{F}}$, writing $\phi_{\mathcal{F}}(\mathbf{x}) = \alpha(\mathbf{x}) \in \mathbb{R}^d$, we have $\mathbb{E}[\alpha(x)\alpha(x)^\top] = I_d$; (3) the loss function $\mathcal{L}(\cdot, \cdot)$ is L-Lipschitz in its first argument and bounded by $B$. Then for any $\delta \in (0, 1)$, with probability at least $1 - \delta$ over $n$ i.i.d. samples $\{(\mathbf{x}_i, \mathbf{y}_i)\}_{i=1}^{n} \sim \mathcal{D}_f$, the following holds simultaneously for all $\omega_f$:*

$$\mathbb{E}_{(\mathbf{x}, \mathbf{y}) \sim \mathcal{D}_f}\big[\mathcal{L}(\hat{y}, \mathbf{y})\big] \leq \frac{1}{n} \sum_{i=1}^{n} \mathcal{L}(\hat{y}_i, \mathbf{y}_i) + 2L\sqrt{\frac{d}{n}} + B\sqrt{\frac{\log(1/\delta)}{2n}}. \tag{7}$$

### 3.2 ICL under Dual-Space Modeling Framework

In this section, we specialize our modeling framework to the ICL setting, with the goal of formalizing the conflation in how Transformers encode context and samples. We first define the task representation space in ICL, which is induced from the context.

**Definition 3.8** (Context-induced task representation in ICL). *In the ICL setting, the task representation can be specified jointly by two components: (1) a context of labeled examples $\mathbf{z}_{1:n} = (\mathbf{z}_1, \ldots, \mathbf{z}_n)$ with $\mathbf{z}_i = (\mathbf{x}_i, \mathbf{y}_i) \in \mathcal{X} \times \mathcal{Y}$, and (2) a representation mapping $\phi : \mathcal{X} \to \mathbb{R}^d$. That is*

$$\omega_f \triangleq \omega_f(\mathbf{z}_{1:n}, \phi). \tag{8}$$

Definition 3.8 formalizes the idea that, in the ICL setting, the task specified by a prompt is determined by its context portion. Thus, in our dual-space framework, the encoding space of context serves as the task representation space. We further show that existing theoretical analyses of ICL based on LSA architectures are fully compatible with our proposed framework, from which we can derive a closed form of $\omega_f$.

**Proposition 3.9** (Closed form of $\omega_f$ under simplified LSA). *Consider an LSA layer applied after a feature encoder $\phi : \mathcal{X} \to \mathbb{R}^d$ implemented by an MLP. Suppose the LSA projection matrices $W_{KQ}$ and $W_{OA}$ are initialized such that*

$$W_{OV} = \begin{pmatrix} * & * \\ 0_d^\top & 1 \end{pmatrix}, \qquad W_{KQ} = \begin{pmatrix} \Theta & 0_d \\ 0_d^\top & * \end{pmatrix}.$$

*Then the final prediction takes the form $\hat{\mathbf{y}} = \langle \omega_f(\mathbf{z}_{1:n}, \phi), \phi(\mathbf{x}_q) \rangle$, where*

$$\omega_f(\mathbf{z}_{1:n}, \phi) = \frac{1}{n} \sum_{i=1}^{n} \mathbf{y}_i \Theta^\top \phi(\mathbf{x}_i). \tag{9}$$

Proposition 3.9 can explain the effectiveness of LSA simplification in analyzing ICL: it implicitly performs our dual-space modeling between the task representation space and the sample representation space. However, we argue that it fails to capture the entanglement of standard Transformers encoding progress, which use the original, unsimplified SA. As we will show in the next theorem, SA cannot realize such dual-space modeling.

**Theorem 3.10** (Entangled structure under general SA). *For a standard SA model with softmax-based attention weights, there does NOT exist a pair of $\phi_0$ and $\omega_0(\mathbf{z}_{1:n}, \phi_0)$, such that the model prediction admits the following decomposition:*

$$\hat{\mathbf{y}}_q = \langle \omega_0(\mathbf{z}_{1:n}, \phi_0), \phi_0(\mathbf{x}_q) \rangle. \tag{10}$$

From our dual-space modeling perspective, Theorem 3.10 formalizes the entangled nature of how Transformers encode context and sample-level information. We posit that this entanglement is the underlying reason for the observed conflict between ICL and IWL.

## 4 CoQE: A Transformer with Separate Context-Query Encoding

We have formalized the entangled nature of standard Transformers encoding progress through a dual-space modeling framework. To address this limitation, we propose a straightforward yet effective architectural modification: CoQE, a Transformer with separate **Co**ntext-**Q**uery **E**ncoding.

The main idea behind CoQE is to disentangle the encoding of context and query: one dedicated to learning in the task representation space and the other to learning in the sample representation space. The CoQE model thus consists of two modules: a shared sample encoder ($\mathcal{E}_{\text{sample}}$) and a dedicated task encoder ($\mathcal{E}_{\text{task}}$), as shown in Figure 1 (b). The sample encoder generates general-purpose representations for all samples, including the query. We implement it with a token-wise module, for it should process samples independently without considering context. The task encoder, on the other hand, operates on the general representations of the context and focuses on producing the representation of the current task. Thus this module should be contextual and has the capability to condense sequential information. Finally, the prediction output is obtained by computing the inner product between the task representation and the query sample representation. Taking the regression task as an example, the formalization of CoQE output is as follows:

$$\hat{\mathbf{y}}_q = \langle \mathcal{E}_{\text{task}}(\mathcal{E}_{\text{sample}}(\mathbf{z}_{1:n})), \mathcal{E}_{\text{sample}}(\mathbf{x}_q) \rangle. \tag{11}$$

Figure 1 compares the architectures of the Transformer and CoQE. The Transformer also contains token-wise components like feed-forward networks, and contextual components like multi-head attention modules. When stacked, these modules collectively exhibit contextual behavior, and the final token output intertwines with the context information in a complex manner during the forward pass. In contrast, CoQE explicitly separates the contextual and token-wise parts, which are responsible for learning the task representation space and the sample representation space, respectively. The two spaces interact through a well-defined inner product according to the Riesz representation theorem.

We aim to evaluate our model across regression and few-shot classification tasks. In the following, we will give the specific implementation of CoQE under both types of tasks. Notably, due to their different properties, the task encoder constructs the task representation space in distinct ways.

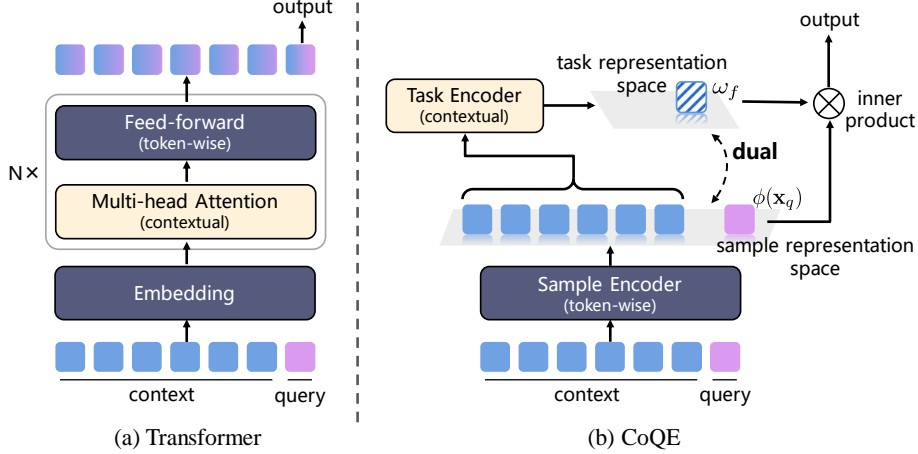

(a) Transformer  (b) CoQE

Figure 1: Comparison of Transformer and CoQE architectures.

## 4.1 IMPLEMENTATION FOR REGRESSION.

We employ a two-layer ReLU network as the sample encoder of CoQE, and a GPT-2–style Transformer as the task encoder. We take the final output token of the task encoder directly as the task representation induced by the context. The regression output is then computed as the inner product between it and the query sample representation. For fair comparison, the baseline Transformer is also equipped with the same two-layer ReLU embedding module.

## 4.2 IMPLEMENTATION FOR FEW-SHOT CLASSIFICATION.

We use a ResNet to encode images input (Chan et al., 2022), which naturally serves as CoQE's sample encoder. We set the embedding dimension of the ResNet to $512$, ensuring sufficient expressiveness. A fully connected layer follows the ResNet to reduce the token dimension back to $64$. The task encoder remains a Transformer, while it constructs the task representation space in a distinct way from regression. A multi-class classification task can be regarded as a collection of sub-tasks that identify each class. Thus, we let it correspond to a set of task representations, each of which is associated with one class. ICL requires producing the task representations corresponding to the classes in the context, whereas IWL requires static memorization of all classes. To construct a task representation

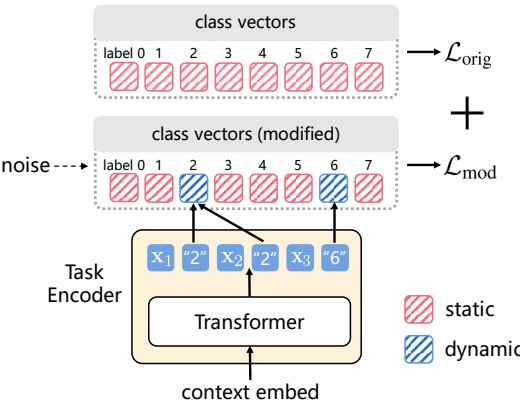

Figure 2: task representation space construction on few-shot classification.

set compatible with both, we assign a parameterized vector to each class, representing a static version of its task representation.

In each forward pass, the classes appearing in the context are encoded by the task encoder to obtain their corresponding task representations, as illustrated in Figure 2. These dynamic vectors replace the corresponding static class vectors, and modified class vectors are used to compute logits for prediction. The resulting training loss is denoted as $\mathcal{L}_{\text{mod}}$. Additionally, to accelerate the training of the static class vectors, we compute an additional set of logits from the unmodified class vectors during training, with the resulting classification loss denoted as $\mathcal{L}_{\text{orig}}$. These logits are not used during testing. Therefore, the total training loss is $\mathcal{L}_{\text{mod}} + \mathcal{L}_{\text{orig}}$.

During experiments, we observed that $\mathcal{L}_{\text{mod}}$ tends to converge to $\mathcal{L}_{\text{orig}}$, which means the task encoder fails to dynamically encode the context over training, and the learning of the task representation space is restricted to the set of static class vectors. It again reflects that the ICL strategy is transient and prone to collapsing into a more stable one, i.e., IWL. To prevent this, we add Gaussian noise to the modified logits during training, with the variance increasing over training steps. The initial

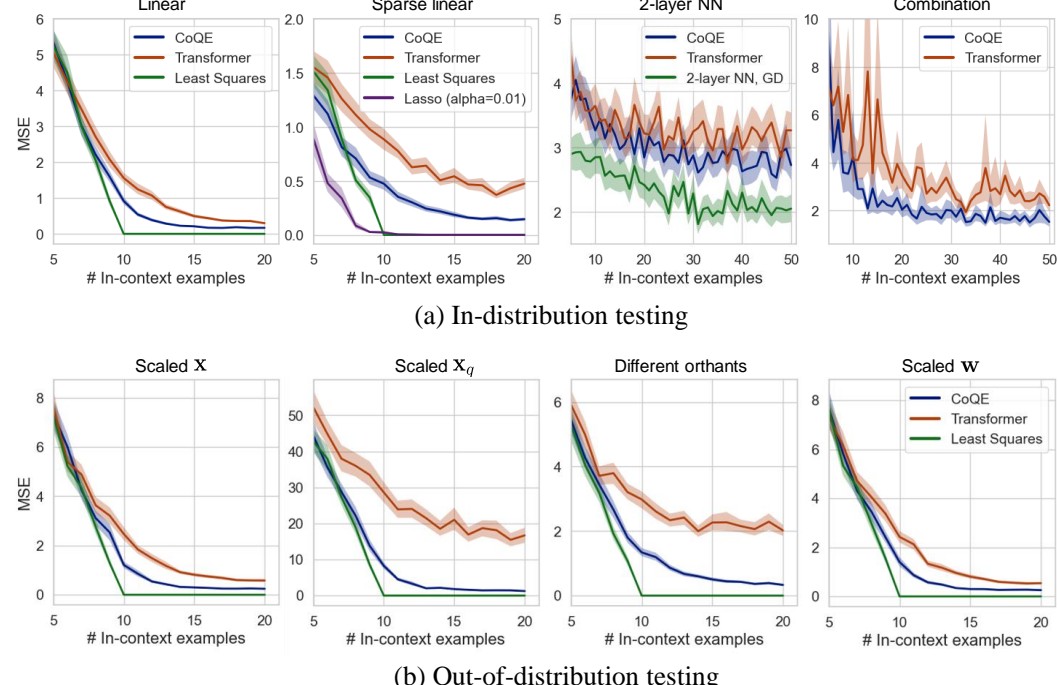

(a) In-distribution testing

(b) Out-of-distribution testing

Figure 3: Results of regression. We provide optimal baselines for most evaluation settings.

noise follows $\mathcal{N}(\mu_0, 1)$. This trick can be interpreted as indirectly performing random sampling in the task representation space. Experimental results and further ablation studies are presented later.

## 5 EXPERIMENTS

In this section, we evaluate the ICL capability, as well as the ICL-IWL compatibility of CoQE across regression and classification tasks. Additional experimental details are provided in Appendix C.

### 5.1 REGRESSION

**Setup.** We adopt a general framework for training models to perform ICL over a function class $\mathcal{F}$. To construct training prompts, we first sample a task function $f \sim \mathcal{D}_{\mathcal{F}}^{\text{train}}$, then draw $k$ i.i.d. inputs $\mathbf{x}_1, \ldots, \mathbf{x}_k \sim \mathcal{D}_{\mathcal{X}}^{\text{train}}$. The prompt is formed as $\mathcal{P} = (\mathbf{x}_1, f(\mathbf{x}_1), \ldots, \mathbf{x}_k, f(\mathbf{x}_k))$. Let $\mathcal{P}^i$ denotes the prefix containing the first $i$ input-output examples and the $(i+1)$th input: $\mathcal{P}^i = (\mathbf{x}_1, f(\mathbf{x}_1), \ldots, \mathbf{x}_i, f(\mathbf{x}_i), \mathbf{x}_{i+1})$. The training objective of a model $\mathbb{M}_\theta$ minimizes the expected loss over all possible prefixes:

$$\min_\theta \ \mathbb{E}_{\mathcal{P}} \left[ \frac{1}{k} \sum_{i=0}^{k-1} \ell\big(\mathbb{M}_\theta(\mathcal{P}^i), \ f(\mathbf{x}_{i+1})\big) \right],$$

where $\ell(\cdot, \cdot)$ is a mean squared error (MSE) loss function. At test time, we first sample a test function $f \sim \mathcal{D}_{\mathcal{F}}^{\text{test}}$, then draw $j \leq k-1$ inputs $\mathbf{x}_1, \ldots, \mathbf{x}_j \sim \mathcal{D}_{\mathcal{X}}^{\text{test}}$, and $\mathbf{x}_q$ from $\mathcal{D}_{\text{query}}$ to construct the test prompt: $\mathcal{P}_{\text{test}}^j = (\mathbf{x}_1, f(\mathbf{x}_1), \ldots, \mathbf{x}_j, f(\mathbf{x}_j), \mathbf{x}_q)$. We evaluate performance still by measuring the MSE between $\mathbb{M}_\theta(\mathcal{P}_{\text{test}}^j)$ and $f(\mathbf{x}_q)$.

To compare our CoQE with the standard Transformer, we consider two major evaluation scenarios: in-distribution (ID) testing and out-of-distribution (OOD) testing. For ID testing, we set $\mathcal{D}_{\mathcal{X}}^{\text{train}} = \mathcal{D}_{\mathcal{X}}^{\text{test}} = \mathcal{D}_{\text{query}}$, and $\mathcal{D}_{\mathcal{F}}^{\text{train}} = \mathcal{D}_{\mathcal{F}}^{\text{test}}$. Specifically, we use the following four classes of functions $\mathcal{F}$: linear functions, sparse linear functions, two-layer ReLU networks and combination functions. The latter two classes of nonlinear functions allow the model to reduce ICL difficulty by learning task-invariant representations. Through them, we can empirically validate Theorem 3.6, which shows the benefits of dual-space modeling for representation learning. For OOD testing, we consider four different cases of distribution shifts under linear functions. See Appendix C.1 for more setup details.

**Results.** In the ID scenario, CoQE consistently achieves lower ICL error than the Transformer (Figure 3 (a)). For regression on more challenging combination functions, the Transformer exhibits substantial fluctuations, whereas CoQE attains much smaller error variance. We attribute this to CoQE's more effective learning of the sample representation space, and present further results in Appendix C.1. In the OOD scenario, CoQE also achieves substantially lower error than the Transformer across all four tested cases (Figure 3 (b)). Notably, the second case is adapted from Mittal et al. (2025), who similarly aims to enforce the model to explicitly learn task variables. However, they found no improvement in OOD performance, contrary to our results. This indicates that simply introducing task variables is insufficient and highlights the value of our proposed dual-space modeling and corresponding architecture design.

## 5.2 FEW-SHOT CLASSIFICATION

**Setup.** To evaluate ICL and IWL abilities under various conditions, we construct prompt sequences that each consists of eight image-label pairs followed by a query image (Chan et al., 2022). The training objective minimizes the cross-entropy loss between the model's prediction and the correct label for the query image.

Training sequences have two key properties that affect the tradeoff between ICL and IWL: burstiness and Zipfian exponent. In bursty sequences, three out of the eight image-label pairs in the context share the same class as the query sample. This setup allows the model to infer the correct label based on context alone, which has been found to incentivize ICL while suppressing IWL (Chan et al., 2022). To avoid repetition biases, bursty sequences additionally include three image-label pairs from a distinct distractor class. $P_{\text{bursty}}$ denotes the proportion

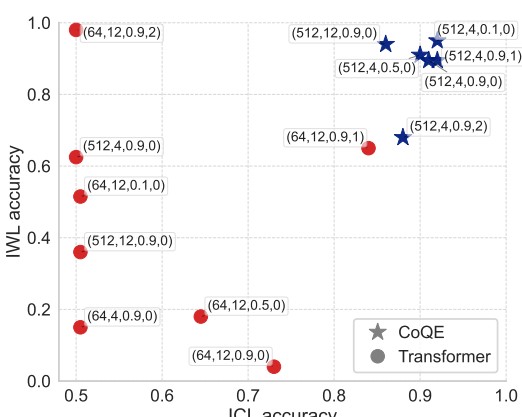

Figure 4: Results under different settings of factors. The annotations in the figure indicate the settings: $(E, L, P_{\text{bursty}}, \alpha)$.

of bursty sequences in the training set, while the rest are generated via random sampling. The second factor is the Zipfian exponent, which controls the frequency distribution of different classes. Under the Zipfian distribution, the class probability is defined as $p(R = r) \propto 1/r^{\alpha}$, where $R$ is the rank of the class, and $\alpha$ is the Zipfian exponent. When $\alpha = 0$, the distribution becomes uniform. Chan et al. (2022) observe that when $\alpha = 1$, a *sweet spot* emerges, where the model reaches a tradeoff for both ICL and IWL. During training, we keep image-label mappings fixed.

Test sequences are divided into two kinds, corresponding respectively to the evaluation of ICL and IWL capability. For ICL, we use sequences with four images from each of two classes unseen in training, and we set the class labels to either 0 or 1 randomly for each sequence. Accuracy on this evaluator is measured across 0 and 1 as possible outputs, and chance-level accuracy is 50%. As these labels are not associated with these images during training, the only way to achieve above-chance accuracy is to refer back to the context. For IWL, we use sequences where none of the context images come from the same class as the query, but all of the image-label mappings are the same as during training. In this case, ICL is not useful, as there are no matching images in context, so the model must rely on mappings stored in weights. See Appendix C.2 for more setup details.

**Model size also affects the ICL-IWL tradeoff.** Before evaluating the algorithmic performance, we make a new finding that model size also strongly affects the ICL-IWL tradeoff in standard Transformers, beyond data distribution factors like burstiness and Zipfian exponent. Specifically, we examine the number of Transformer layers $L$ and the embedding dimension of the ResNet $E$. We observe that, under the same conditions, a 12-layer Transformer exhibits stronger ICL but weaker IWL compared to a 4-layer Transformer. We suppose that this is due to the Transformer's inductive bias toward attending to context, compared to just memorizing context-irrelevant sample information. Another interesting finding is that increasing the ResNet embedding dimension from 64 to 512 nearly eliminates the model's ICL ability while substantially enhancing IWL. Notably, we connect a fully connected layer after the ResNet to reduce the dimension back to 64 before inputting to the

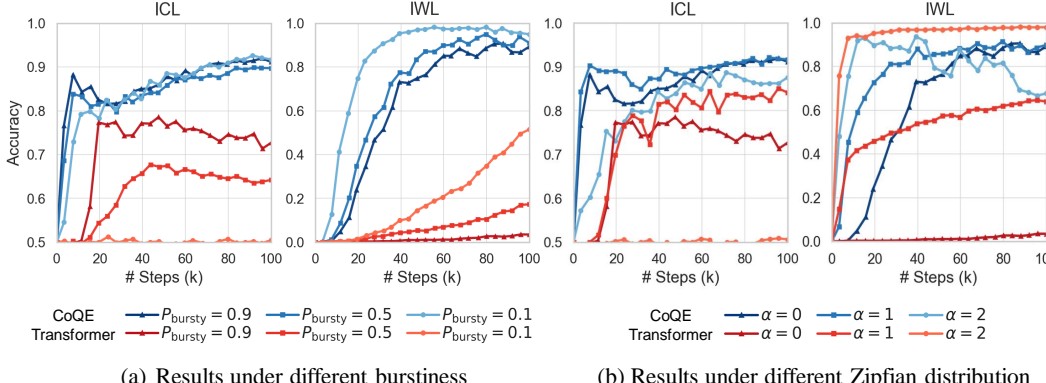

(a) Results under different burstiness      (b) Results under different Zipfian distribution

Figure 5: Learning curves under different data distribution factors.

Transformer, ensuring that the latter's role remains unchanged. We speculate that the larger ResNet increases the expressivity of individual tokens, and when a single token is sufficiently expressive to solve the task, the model tends to ignore the context. This is consistent with Singh et al. (2023)'s observation that applying $\ell_2$ regularization to the ResNet can bias the tradeoff toward ICL. Our finding further highlights the complex intertwining between ICL and IWL in standard Transformers.

**Results.** Figure 4 presents the ICL and IWL accuracies of Transformers and CoQE under various factors after 100k training steps. The Transformers fluctuate between ICL and IWL capabilities across different conditions, whereas our models robustly occupy the upper-right region, indicating a Pareto improvement in both abilities. Figure 5 shows the learning curves under different values of $P_{\text{bursty}}$ and Zipfian exponent. We could observe that CoQE's ICL accuracy rises rapidly at the beginning, but declines slightly between 10k and 30k steps. This behavior aligns with prior findings on ICL strategy: it emerges quickly and then gradually fades (Singh et al., 2023). However, under our algorithm, the model quickly restrains this fading trend and continues to recover steadily. We also discuss the issue of parameter scale, as presented in Appendix C.2.

**Ablation study.** We study the effect of Gaussian noise on the model performance, as shown in Table 1. Without any noise, the model's ICL ability ultimately yields entirely to IWL. When $\mu_0 = 5$, the model achieves maximal ICL performance while retaining high IWL capability. This is the default noise magnitude used in our experiments. See Appendix C.2 for more details.

Table 1: Results under different noise levels.

|  | ICL | IWL |
|---|---|---|
| Noise-free | 55.12 | 99.62 |
| $\mu_0 = 3$ | 81.91 | 95.31 |
| $\mu_0 = 5$ | 91.15 | 89.30 |
| $\mu_0 = 7$ | 88.22 | 77.70 |
| $\mu_0 = 9$ | 86.01 | 72.62 |

## 6 DISCUSSION

In this section, we briefly discuss three issues of concern. **Firstly, why do large language models not exhibit a clear imbalance between ICL and IWL?** Piantadosi (2014) showed that a Zipfian distribution of $\alpha = 1$ closely approximates the empirical distribution of natural language, which serves as a sweet spot for the tradeoff between ICL and IWL (Chan et al., 2022). On the other hand, Chan et al. (2025) pointed out that LLMs still face conflicts between ICL and IWL in some scenarios. **Secondly, why is it important to reconcile ICL and IWL under diverse conditions?** Because with the growing demand of multimodal large models (e.g., VLMs, VLAs) for increasingly diverse data distributions, as well as the emergence of new model architectures, relying on the fortunate coincidence of natural language data distributions is far from sufficient to ensure robust performance. **Thirdly, how can our algorithm scale to larger models and other tasks?** The core of our method can be abstracted as blockwise processing of the input sequence into a context part and a query part, thereby learning two spaces of different semantic significance. In more general scenarios, the query may not be limited to the last token but could instead be the user's explicit question (Chen et al., 2025; Zong et al., 2025). Therefore, the notion of a sample may also need to go beyond a single token and be redefined as a sequence-level sample, which we leave for future work. For typical ICL scenarios where the context provides concrete demonstration examples, we argue that our algorithm could facilitate the model's ICL performance. When the context consists of more general information such as historical cues or task instructions, it can still be beneficial by helping the model distill relevant information. For pretrained LLMs under the current architecture, our algorithm can be implemented by adding an auxiliary branch for context processing.

## ETHICS STATEMENT

Our study does not involve human participants, sensitive data, or any foreseeable negative societal impact. Hence, it does not raise ethical concerns related to the ICLR Code of Ethics.

## REPRODUCIBILITY STATEMENT

We have taken measures to ensure the reproducibility of our work. Theoretical results are accompanied by detailed proofs in Appendix B. Experimental setups and implementation details are fully described in Section 5 and Appendix C. We also provide the source code in the supplementary material, which is based on the repositories of Garg et al. (2022) and Chan et al. (2022).

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

# A RELATED WORK

**Theoretical Investigations on ICL Mechanisms.** Recent theoretical work has examined how Transformers perform ICL across various scenarios (Zhang et al., 2024a; Li et al., 2023; Tian et al., 2023; Nichani et al., 2024; Chen et al., 2024; Wu et al., 2024; Huang & Ge, 2024; Oko et al., 2024; Liang et al., 2025). These studies typically analyze simplified architectures such as linear self-attention or query-key-combined formulations. Bu et al. (2025) extends the theoretical analysis to nonlinear transformers incorporating LayerNorm, though retaining linear self-attention mechanisms. Some other studies (Zhang et al., 2025b; Ye et al., 2024) conduct analyses from the perspective of Bayesian model averaging, but they likewise rely on unrealistic assumptions that distort Transformer architectures for kernel regression. In this paper, we demonstrate the validity of the linear attention simplification as a special case consistent with our dual-space modeling, while also showing that the standard softmax self-attention does not support such modeling. The latter serves as the starting point for our improved architectural design.

**Empirical Investigations on ICL Mechanisms.** Garg et al. (2022) firstly demonstrated that Transformer-based ICL can generalize effectively to out-of-distribution (OOD) tasks, leading to a surge of interest in exploring its generalization behavior (Ahuja & Lopez-Paz, 2023; Kossen et al., 2024; Pan et al., 2023; Fan et al., 2024). Xiong et al. (2025) showed that LLMs can perform different ICL functions during a single inference, while Yadlowsky et al. (2023) and Wang et al. (2025) revealed that Transformers often face challenges when generalizing to unseen functions. Another line of studies focuses on the function reference capability of Transformers underlying their ICL performance. Some work has shown that LLM can implicitly encode task vectors during ICL (Hendel et al., 2023; Todd et al., 2024; Guo et al., 2024; Yang et al., 2025; Han et al., 2025). Mittal et al. (2025) enforced the explicit task variables learning by introducing a bottleneck to the Transformer, yet found no improvement in OOD performance of ICL, contrary to our results. This indicates that simply introducing task variables is insufficient and highlights the value of our proposed modeling of task representation space, along with corresponding architecture design.

**Relationship between ICL and IWL.** Beyond investigations on the ICL mechanisms, some studies have found that ICL is not a guaranteed and stable capability of Transformers; rather, it competes with the model's inherent in-weight learning (IWL) ability, which relies on information stored in the weights (Chan et al., 2022; Singh et al., 2023; Reddy, 2024; Panwar et al., 2024). Chan et al. (2022) examined the impact of different training data distributions on both abilities, finding that burstiness and skewed distributions significantly affect their tradeoff. Only when the training data follows a certain distribution can both abilities coexist. Singh et al. (2023) further confirmed the transient nature of ICL, observing that it always fades after emerging and gives way to IWL. They hypothesize that this phenomenon arises from the competition between the two strategies for the shared model circuits. Nguyen & Reddy (2025) on the other hand, attributes this to the different relative learning rates of ICL and IWL, and conducted an analysis on a simplified one-layer transformer model. Chan et al. (2025) proposed a simple theoretical model, which is a linear combination of an in-weight learner and an in-context learner. Singh et al. (2025) empirically discovered a more complex coopetition relationship between ICL and IWL. However, to date, no work has truly resolved the challenge of achieving robust coexistence between ICL and IWL.

**Linearization in Latent Space.** Beyond task-specific vectors, a line of work has examined how large models internally encode a variety of abstract concepts as linear vectors in latent space, giving rise to the commonly accepted *linear representation hypothesis* (Mikolov et al., 2013; Nanda et al., 2023; Park et al., 2024). Several studies have shown that concepts such as truthfulness (Marks & Tegmark, 2024), time and space (Gurnee & Tegmark, 2024), and other semantic properties (Dalvi et al., 2022; Merullo et al., 2024; Ye et al., 2025) can emerge in the model's latent space, using linear probes as the primary tool. Additionally, larger models tend to yield more disentangled and interpretable internal representations (Bricken et al., 2023; Cunningham et al., 2023), and this can be regarded as evidence of the emergence of a world model within large scale networks (Zhang et al., 2025a). In this work, we propose the concept of a linear task representation space, grounded in the linear representational hypothesis. This modeling aligns with empirical observations of task vectors, and further serves as a theoretical extension and utilization of linearization in the model's latent space.

# B   PROOFS OF THEORETICAL RESULTS

## B.1   PROOF OF PROPOSITION 3.3

For ease of presentation, we first restate the proposition and then introduce its proof.

**Proposition B.1** (Task-sample duality). *Let $\mathcal{X}$ be the input space and $\mathcal{Y}_f$ the multiple label sets corresponding to each task $f \in \mathcal{F}$. Under Definition 3.2, there exists a linear sample representation space $\mathcal{M}_{\mathcal{F}}$ and a linear task transformation space $\mathcal{T}$, where $\mathcal{T}$ is the dual space of $\mathcal{M}_{\mathcal{F}}$, i.e. $\mathcal{T} = \mathcal{M}_{\mathcal{F}}^*$.*

*Proof.* To prove the proposition, we must show that the linear task transformation space $\mathcal{T}$ is equivalent to the dual space of the linear sample representation space $\mathcal{M}_{\mathcal{F}}$, denoted as $\mathcal{M}_{\mathcal{F}}^*$. The proof proceeds by demonstrating mutual inclusion: (1) $\mathcal{T} \subseteq \mathcal{M}_{\mathcal{F}}^*$ and (2) $\mathcal{M}_{\mathcal{F}}^* \subseteq \mathcal{T}$.

**Step 1: Proof of $\mathcal{T} \subseteq \mathcal{M}_{\mathcal{F}}^*$.**   Let $t$ be an arbitrary element in the task transformation space $\mathcal{T}$. According to Definition 3.2, $t$ is a linear function such that

$$t(m) = \langle \omega_t, m \rangle, \quad \forall m \in \mathcal{M}_{\mathcal{F}}.$$

Since $t$ is a linear functional on $M_{\mathcal{F}}$, it is by definition an element of $M_{\mathcal{F}}^*$. As $t$ was an arbitrary element of $\mathcal{T}$, it follows that every element in $\mathcal{T}$ corresponds to a unique linear functional in $M_{\mathcal{F}}^*$. Thus, we have established that $\mathcal{T} \subseteq \mathcal{M}_{\mathcal{F}}^*$.

**Step 2: Proof of $\mathcal{M}_{\mathcal{F}}^* \subseteq \mathcal{T}$.**   Conversely, let $t'$ be an arbitrary linear functional in the dual space $\mathcal{M}_{\mathcal{F}}^*$. By Definition 3.1, $\mathcal{M}_{\mathcal{F}}$ is a finite-dimensional inner product space. By the Riesz representation theorem, for any linear functional $t' \in \mathcal{M}_{\mathcal{F}}^*$, there exists a unique vector, let's call it $\omega_{t'} \in \mathcal{M}_{\mathcal{F}}$, such that for all $m \in \mathcal{M}_{\mathcal{F}}$:

$$t'(m) = \langle \omega_{t'}, m \rangle.$$

Now, let us define a function $f_{t'} : \mathcal{X} \to \mathbb{R}$ using this functional $t'$:

$$f_{t'}(\mathbf{x}) = t'(\phi_{\mathcal{F}}(\mathbf{x})) = \langle \omega_{t'}, \phi_{\mathcal{F}}(\mathbf{x}) \rangle.$$

This function $f_{t'}$ has the exact mathematical form of a task function as specified in Definition 3.2. Therefore, $f_{t'}$ can be considered a valid task belonging to the task function space $\mathcal{F}$. Definition 3.2 states that for any such task $f_{t'} \in \mathcal{F}$, there exists a unique linear task representation vector, which we denote $\omega_f$, that represents it. This means:

$$f_{t'}(\mathbf{x}) = \langle \omega_f, \phi_{\mathcal{F}}(\mathbf{x}) \rangle.$$

By equating the two expressions for $f_{t'}(\mathbf{x})$, we obtain:

$$\langle \omega_{t'}, \phi_{\mathcal{F}}(\mathbf{x}) \rangle = \langle \omega_f, \phi_{\mathcal{F}}(\mathbf{x}) \rangle, \quad \forall \mathbf{x} \in \mathcal{X}$$

This implies that $\langle \omega_{t'} - \omega_f, m \rangle = 0$ for all $m$ in the image of $\phi_{\mathcal{F}}$. Since the sample representation space $\mathcal{M}_{\mathcal{F}}$ is spanned by the image of $\phi_{\mathcal{F}}$, this condition holds for all $m \in \mathcal{M}_{\mathcal{F}}$. The only vector orthogonal to every vector in an inner product space is the zero vector. Therefore:

$$\langle \omega_{t'} - \omega_f, m \rangle = 0 \implies \omega_{t'} = \omega_f.$$

Since $\omega_f$ corresponds to an element of $\mathcal{T}$, it follows that $\omega_{t'}$ is also corresponds to an element of $\mathcal{T}$, and thus $t' \in \mathcal{T}$. As our choice of $t'$ was arbitrary, we have shown that every linear functional in $\mathcal{M}_{\mathcal{F}}^*$ corresponds to an element in $\mathcal{T}$. Thus, we have established that $\mathcal{M}_{\mathcal{F}}^* \subseteq \mathcal{T}$.   $\square$

## B.2   PROOF OF THEOREM 3.6

For ease of presentation, we first restate the theorem and then introduce its proof.

**Theorem B.2** (Completeness of basis representations under task traversal). *Under Proposition 3.3, we assume that a learner with sample representation mapping $\phi_\theta$ is presented with a task traversal curriculum $\mathcal{C}$ such that: $\mathrm{span}\{t \mid t \in \mathcal{C}\} = \mathcal{T}$. Then, if the learner achieves zero empirical error, the learned representation mapping $\phi_\theta$ satisfies: $\mathrm{span}\{\phi_\theta(\mathbf{x}) \mid \mathbf{x} \in \mathcal{X}\} = \mathcal{M}_{\mathcal{F}}$; equivalently, each basis sample representation $m_i$ occurs in $\phi_\theta$.*

*Proof.* By Proposition 3.3, fix a basis $\{m_i\}_{i=1}^d$ of $\mathcal{M}_\mathcal{F}$ and its dual basis $\{t_i\}_{i=1}^d \subset \mathcal{T}$, satisfying $t_i(m_j) = \langle \omega_i, m_j \rangle = \delta_{ij}$. For any $m \in \mathcal{M}_\mathcal{F}$ write the unique decomposition $m = \sum_{i=1}^d \alpha_i(m)\, m_i$ and for any $t \in \mathcal{T}$ write $t = \sum_{i=1}^d \beta_i(t)\, t_i$. The bilinear pairing then reduces to

$$t(m) = \sum_{i=1}^d \alpha_i(m)\, \beta_i(t). \tag{12}$$

Let $\phi_\mathcal{F} : \mathcal{X} \to \mathcal{M}_\mathcal{F}$ denote the sample representation guaranteed by Definition 3.2, and define the coordinate vectors

$$\alpha^\theta(\mathbf{x}) \triangleq (\alpha_1(\phi_\theta(\mathbf{x})), \ldots, \alpha_d(\phi_\theta(\mathbf{x}))) \in \mathbb{R}^d, \qquad \alpha^*(\mathbf{x}) \triangleq (\alpha_1(\phi_\mathcal{F}(\mathbf{x})), \ldots, \alpha_d(\phi_\mathcal{F}(\mathbf{x}))) \in \mathbb{R}^d.$$

Zero empirical error on every curriculum task $t \in \mathcal{C}$ means

$$t(\phi_\theta(\mathbf{x})) = t(\phi_\mathcal{F}(\mathbf{x})) \quad \text{for all } t \in \mathcal{C} \text{ and all training } \mathbf{x}.$$

By linearity of Equation 12 this equality holds for any linear combination of curriculum tasks; hence it holds for all $t \in \mathrm{span}(\mathcal{C}) = \mathcal{T}$:

$$t(\phi_\theta(\mathbf{x})) = t(\phi_\mathcal{F}(\mathbf{x})), \qquad \forall t \in \mathcal{T}. \tag{13}$$

Take in Equation 13 the particular choice $t = t_i$ (the $i$-th dual basis functional). Using $t_i(m) = \alpha_i(m)$ from Equation 12, we obtain for every $\mathbf{x}$ and every $i \in [d]$,

$$\alpha_i(\phi_\theta(\mathbf{x})) = t_i(\phi_\theta(\mathbf{x})) = t_i(\phi_\mathcal{F}(\mathbf{x})) = \alpha_i(\phi_\mathcal{F}(\mathbf{x})).$$

Thus $\alpha^\theta(\mathbf{x}) = \alpha^*(\mathbf{x})$ pointwise for all (training) $\mathbf{x}$. Consequently $\phi_\theta(\mathbf{x})$ and $\phi_\mathcal{F}(\mathbf{x})$ have identical coordinates in the basis $\{m_i\}_{i=1}^d$ for all $\mathbf{x}$, so

$$\mathrm{span}\{\phi_\theta(\mathbf{x}) \mid \mathbf{x} \in \mathcal{X}\} = \mathrm{span}\{\phi_\mathcal{F}(\mathbf{x}) \mid \mathbf{x} \in \mathcal{X}\}.$$

Without loss of generality, take $\mathcal{M}_\mathcal{F} = \mathrm{span}\{\phi_\mathcal{F}(\mathbf{x}) \mid \mathbf{x} \in \mathcal{X}\}$. Therefore $\mathrm{span}\{\phi_\theta(\mathbf{x}) \mid \mathbf{x} \in \mathcal{X}\} = \mathcal{M}_\mathcal{F}$, proving the first claim.

Finally, since $\{m_i\}_{i=1}^d$ is a basis of $\mathcal{M}_\mathcal{F}$, for each $i$ there exists some $\mathbf{x}$ with $\alpha_i(\phi_\mathcal{F}(\mathbf{x})) \neq 0$; by the coordinate equality above, $\alpha_i(\phi_\theta(\mathbf{x})) \neq 0$ for the same $\mathbf{x}$. Hence each basis sample representation $m_i$ occurs in $\phi_\theta$. $\qquad\square$

### B.3 PROOF OF THEOREM 3.7

For ease of presentation, we first restate the theorem and then introduce its proof.

**Theorem B.3** (Generalization error bound). *Under Proposition 3.3 and Definition 3.4, for any task $f$ represented by $\omega_f$ and input $\mathbf{x}$ represented by $\phi_\mathcal{F}(\mathbf{x})$, the predictor is $\hat{y} = \langle \omega_f, \phi_\mathcal{F}(\mathbf{x}) \rangle$. We assume that (1) $\|\omega_f\|_2 \leq 1, \forall f \in \mathcal{F}$; (2) the feature map is isotropic: for an orthonormal basis $\{m_j\}_{j=1}^d$ of $\mathcal{M}_\mathcal{F}$, writing $\phi_\mathcal{F}(\mathbf{x}) = \alpha(\mathbf{x}) \in \mathbb{R}^d$, we have $\mathbb{E}[\alpha(x)\alpha(x)^\top] = I_d$; (3) The loss function $\mathcal{L}(\cdot, \cdot)$ is L-Lipschitz in its first argument and bounded by $B$. Then for any $\delta \in (0,1)$, with probability at least $1 - \delta$ over $n$ i.i.d. samples $\{(\mathbf{x}_i, \mathbf{y}_i)\}_{i=1}^n \sim \mathcal{D}_f$, the following holds simultaneously for all $\omega_f$:*

$$\mathbb{E}_{(\mathbf{x},\mathbf{y}) \sim \mathcal{D}_f}[\mathcal{L}(\hat{y}, \mathbf{y})] \leq \frac{1}{n} \sum_{i=1}^n \mathcal{L}(\hat{y}_i, \mathbf{y}_i) + 2L\sqrt{\frac{d}{n}} + B\sqrt{\frac{\log(1/\delta)}{2n}}. \tag{14}$$

*Proof.* Let $\mathcal{H} := \{h(\mathbf{x}) = \langle \omega_f, \phi_\mathcal{F}(\mathbf{x}) \rangle : \|\omega_f\|_2 \leq 1\}$ and $\mathcal{G} := \{(\mathbf{x}, \mathbf{y}) \mapsto \mathcal{L}(\hat{y}, \mathbf{y}) : \omega_f \in \mathcal{T}, \|\omega_f\|_2 \leq 1\}$. By the standard uniform deviation bound via (empirical) Rademacher complexity, for any $\delta \in (0,1)$, with probability at least $1 - \delta$ over the sample,

$$\forall g \in \mathcal{G}: \quad \mathbb{E}[g] \leq \frac{1}{n} \sum_{i=1}^n g(x_i, y_i) + 2\hat{\mathfrak{R}}_n(\mathcal{G}) + B\sqrt{\frac{\log(1/\delta)}{2n}}, \tag{15}$$

where $\hat{\mathfrak{R}}_n(\mathcal{G}) := \mathbb{E}_\sigma\left[\sup_{g \in \mathcal{G}} \frac{1}{n} \sum_{i=1}^n \sigma_i g(\mathbf{x}_i, \mathbf{y}_i)\right]$ and $\sigma_i$ are i.i.d. Rademacher signs. By the vector-contraction inequality, because $\mathcal{L}(\cdot, \cdot)$ is $L$–Lipschitz,

$$\hat{\mathfrak{R}}_n(\mathcal{G}) \leq L\, \hat{\mathfrak{R}}_n(\mathcal{H}), \qquad \hat{\mathfrak{R}}_n(\mathcal{H}) := \mathbb{E}_\sigma\left[\sup_{\|\omega_f\| \leq 1} \frac{1}{n} \sum_{i=1}^n \sigma_i \langle \omega_f, \phi_\mathcal{F}(\mathbf{x}_i) \rangle\right]. \tag{16}$$

For any fixed sample $S = \{\mathbf{x}_i\}_{i=1}^n$,

$$\hat{\mathfrak{R}}_n(\mathcal{H}) = \frac{1}{n} \mathbb{E}_\sigma \Big\| \sum_{i=1}^n \sigma_i \phi_{\mathcal{F}}(\mathbf{x}_i) \Big\|_2 \leq \frac{1}{n} \sqrt{\mathbb{E}_\sigma \Big\| \sum_{i=1}^n \sigma_i \phi_{\mathcal{F}}(\mathbf{x}_i) \Big\|_2^2} = \frac{1}{n} \sqrt{\sum_{i=1}^n \|\phi_{\mathcal{F}}(\mathbf{x}_i)\|_2^2}.$$

Writing $\phi_{\mathcal{F}}(\mathbf{x}_i) = \alpha(\mathbf{x}_i) \in \mathbb{R}^d$ in the fixed orthonormal basis $\{m_j\}_{j=1}^d$, we have $\|\phi_{\mathcal{F}}(\mathbf{x}_i)\|_2^2 = \|\alpha(\mathbf{x}_i)\|_2^2$ and, by isotropy, $\mathbb{E}\|\alpha(\mathbf{x})\|_2^2 = \mathrm{tr}\big(\mathbb{E}[\alpha(\mathbf{x})\alpha(\mathbf{x})^\top]\big) = d$. Hence

$$\hat{\mathfrak{R}}_n(\mathcal{H}) \leq \sqrt{\frac{d}{n}}. \tag{17}$$

Combining the Equations 15 16 17 yields that, with probability at least $1 - \delta$, for all $\omega_f$ with $\|\omega_f\| \leq 1$,

$$\mathbb{E}_{(\mathbf{x},\mathbf{y})\sim\mathcal{D}_f}[\mathcal{L}(\hat{\mathbf{y}}, \mathbf{y})] \leq \frac{1}{n} \sum_{i=1}^n \mathcal{L}(\hat{\mathbf{y}}_i, \mathbf{y}_i) + 2L\sqrt{\frac{d}{n}} + B\sqrt{\frac{\log(1/\delta)}{2n}}.$$

This is the desired inequality, and it holds simultaneously for all tasks $f$ (equivalently, all $\omega_f$ with $\|\omega_f\| \leq 1$) by the same uniform bound. $\qquad\square$

## B.4 PROOF OF PROPOSITION 3.9

For ease of presentation, we first restate the proposition and then introduce its proof.

**Proposition B.4** (Closed form of $\omega_f$ under simplified LSA). *Consider an LSA layer applied after a feature encoder $\phi : \mathcal{X} \to \mathbb{R}^d$ implemented by an MLP. Suppose the LSA projection matrices $W_{KQ}$ and $W_{OA}$ are initialized such that*

$$W_{OV} = \begin{pmatrix} * & * \\ 0_d^\top & 1 \end{pmatrix}, \qquad W_{KQ} = \begin{pmatrix} \Theta & 0_d \\ 0_d^\top & * \end{pmatrix}.$$

*Then the final prediction takes the form $\hat{\mathbf{y}} = \langle \omega_f(\mathbf{z}_{1:n}, \phi), \phi(\mathbf{x}_q) \rangle$, where*

$$\omega_f(\mathbf{z}_{1:n}, \phi) = \frac{1}{n} \sum_{i=1}^n \mathbf{y}_i \Theta^\top \phi(\mathbf{x}_i). \tag{18}$$

*Proof.* According to Kim & Suzuki (2024), under the conditions of Proposition 3.9, the expression of $\hat{\mathbf{y}}$ is given as follows:

$$\hat{\mathbf{y}} = \frac{1}{n} \sum_{i=1}^n \mathbf{y}_i \phi(\mathbf{x}_i)^\top \Theta \phi(\mathbf{x}_q). \tag{19}$$

Hence, Proposition 3.9 is readily proved.

$\qquad\square$

## B.5 PROOF OF THEOREM 3.10

For ease of presentation, we first restate the theorem and then introduce its proof.

**Theorem B.5** (Entangled structure under general SA). *For a standard SA model with softmax-based attention weights, there does NOT exist a pair of $\phi_0$ and $\omega_0(\mathbf{z}_{1:n}, \phi_0)$, such that the model prediction admits the following decomposition:*

$$\hat{\mathbf{y}}_q = \langle \omega_0(\mathbf{z}_{1:n}, \phi_0), \phi_0(\mathbf{x}_q) \rangle. \tag{20}$$

*Proof.* We argue by contradiction. Assume there exists a finite-dimensional feature map $\phi_0$ and a context-only coefficient vector $\omega_0(\mathbf{z}_{1:n}, \phi_0)$ such that the identity holds for all contexts and queries.

**Step 1: From SA equations to a ratio of exponentials in $\mathbf{x}_q$.** Let the sequence length be $L = n+1$. Stack token embeddings as $Z = [\mathbf{z}_1, \ldots, \mathbf{z}_n, \mathbf{z}_q] \in \mathbb{R}^{d \times L}$. A single-head self-attention (SA) layer computes

$$Q = W_Q Z, \quad K = W_K Z, \quad V = W_V Z,$$

with $Q, K \in \mathbb{R}^{d_k \times L}$, $V \in \mathbb{R}^{d_v \times L}$. Denote the $i$-th key/value columns by $k_i := K_{:i} = W_K \mathbf{z}_i$, $v_i := V_{:i} = W_V \mathbf{z}_i$, and the query column at position $q$ by $q := Q_{:L} = W_Q \mathbf{z}_q$. The attention weights for the query position form a probability vector $\alpha(q) \in \Delta^n$ with coordinates

$$\alpha_i(q) \;=\; \frac{\exp\left(\langle k_i, q \rangle / \sqrt{d_k}\right)}{\sum_{j=1}^{L} \exp\left(\langle k_j, q \rangle / \sqrt{d_k}\right)}, \qquad i = 1, \ldots, L. \tag{21}$$

In the theoretical analysis of ICL, it is common to set $\mathbf{z}_q = [\mathbf{x}_q, 0]$. Without loss of generality, we assume that the query token embedding depends affinely on the input feature $\mathbf{x}_q \in \mathbb{R}^{d_x}$:

$$\mathbf{z}_q \;=\; E_x \mathbf{x}_q + r_q,$$

where $E_x \in \mathbb{R}^{d \times d_x}$ is a fixed embedding matrix and $r_q \in \mathbb{R}^d$ could collect position encodings and other context-independent parts at position $q$. Then the query vector is also affine in $\mathbf{x}_q$:

$$q \;=\; W_Q \mathbf{z}_q \;=\; W_Q E_x \mathbf{x}_q + W_Q r_q \;=\; U \mathbf{x}_q + u_0,$$

with $U := W_Q E_x \in \mathbb{R}^{d_k \times d_x}$ and $u_0 := W_Q r_q \in \mathbb{R}^{d_k}$. Plugging $q = U \mathbf{x}_q + u_0$ into the logits in Equation 21 yields, for each key $i$,

$$\frac{\langle k_i, q \rangle}{\sqrt{d_k}} = \frac{\langle k_i, U \mathbf{x}_q \rangle}{\sqrt{d_k}} + \frac{\langle k_i, u_0 \rangle}{\sqrt{d_k}} = a_i^\top \mathbf{x}_q + b_i(\mathbf{z}),$$

where we define the (query–input) slope and the (context) offset by

$$a_i \;:=\; \frac{U^\top k_i}{\sqrt{d_k}} \in \mathbb{R}^{d_x}, \qquad b_i(\mathbf{z}) \;:=\; \frac{\langle k_i, u_0 \rangle}{\sqrt{d_k}} \in \mathbb{R}.$$

Hence, for a fixed context $\mathbf{z}_{1:n}$ (which fixes all $k_i$ and $u_0$), the attention weights are *softmax of affine functions of* $\mathbf{x}_q$:

$$\alpha_i(\mathbf{x}_q; \mathbf{z}) \;=\; \frac{\exp\left(a_i^\top \mathbf{x}_q + b_i(\mathbf{z})\right)}{\sum_{j=1}^{L} \exp\left(a_j^\top \mathbf{x}_q + b_j(\mathbf{z})\right)}, \qquad i = 1, \ldots, L. \tag{22}$$

The SA output at the query position is $h_q \;=\; \mathbf{z}_q + W_O \sum_{i=1}^{L} \alpha_i(\mathbf{x}_q; \mathbf{z}) v_i$. For a fixed linear predictor $w \in \mathbb{R}^d$ (or equivalently choosing a fixed output coordinate), the scalar prediction is

$$\hat{\mathbf{y}}_q(\mathbf{x}_q) = w^\top h_q = \underbrace{w^\top \mathbf{z}_q}_{\text{affine in } \mathbf{x}_q} + \sum_{i=1}^{L} \underbrace{\left(w^\top W_O v_i\right)}_{:= \gamma_i(\mathbf{z})} \alpha_i(\mathbf{x}_q; \mathbf{z}). \tag{23}$$

If we choose $w$ orthogonal to $\text{Im}(E_x)$ (always possible unless $E_x = 0$), then $w^\top \mathbf{z}_q = w^\top (E_x \mathbf{x}_q + r_q) = w^\top r_q$ is a context-only constant; denote $c(\mathbf{z}) := w^\top r_q$. With $\gamma(\mathbf{z}) := (\gamma_1(\mathbf{z}), \ldots, \gamma_L(\mathbf{z}))^\top$, Equation 23 simplifies to

$$\hat{\mathbf{y}}_q(\mathbf{x}_q) \;=\; c(\mathbf{z}) \;+\; \gamma(\mathbf{z})^\top \alpha(\mathbf{x}_q; \mathbf{z}), \tag{24}$$

where $\alpha(\cdot; \mathbf{z})$ is given by the ratio-of-exponentials form in Equation 22. This exhibits the claimed dependence of $\hat{\mathbf{y}}_q$ on $\mathbf{x}_q$ through a softmax over affine functions of $\mathbf{x}_q$.

**Step 2: A two-key reduction yields a linearly independent logistic family.** Specialize to $d_x = 1$ and one context keys ($n = 1$) with $a_1 \neq a_2$. Choose $W_O, V$ so that $c(\mathbf{z}) \equiv 0$ and $\gamma_1(\mathbf{z}) = 1, \gamma_2(\mathbf{z}) = 0$. Then Equation 24 reduces to

$$\hat{\mathbf{y}}_q(\mathbf{x}_q) \;=\; \frac{\exp(a_1 \mathbf{x}_q + b_1(\mathbf{z}))}{\exp(a_1 \mathbf{x}_q + b_1(\mathbf{z})) + \exp(a_2 \mathbf{x}_q + b_2(\mathbf{z}))} \;=\; \frac{1}{1 + t(\mathbf{z}) e^{-a \mathbf{x}_q}},$$

where $a := a_1 - a_2 \neq 0$ and $t(z) := \exp\big(b_2(z) - b_1(z)\big) > 0$. As the context varies, $t(z)$ can take arbitrarily many distinct positive values, so SA realizes the one-parameter family of functions

$$\mathcal{F} = \Big\{ f_t(\mathrm{x}) := \frac{1}{1 + te^{-a\mathrm{x}}} \; : \; t > 0 \Big\}.$$

Fix distinct $t_1, \ldots, t_m > 0$. Suppose there exist scalars $\lambda_1, \ldots, \lambda_m$ with $\sum_{i=1}^m \lambda_i f_{t_i}(\mathrm{x}) \equiv 0$ for all $\mathrm{x} \in \mathbb{R}$. Multiplying both sides by $\prod_{i=1}^m (1 + t_i e^{-a\mathrm{x}})$ and letting $s = e^{-a\mathrm{x}}$ gives the polynomial identity

$$\sum_{i=1}^m \lambda_i \prod_{j \neq i} (1 + t_j s) \equiv 0 \quad \text{for all } s > 0.$$

A polynomial that vanishes on an infinite set is identically zero; hence the identity holds for all $s \in \mathbb{R}$. Evaluating at $s = -1/t_k$ yields

$$\lambda_k \prod_{j \neq k} \Big( 1 - \frac{t_j}{t_k} \Big) = 0.$$

Since the $t_i$ are distinct, each product is nonzero, forcing $\lambda_k = 0$ for all $k$. Thus $f_{t_1}, \ldots, f_{t_m}$ are linearly independent. Consequently, the linear span of $\mathcal{F}$ is infinite-dimensional.

**Step 3: Contradiction with any finite-dimensional bilinear decomposition.** If the bilinear decomposition $\hat{y}_q(\mathrm{x}_q) = \langle \omega_0(z), \phi_0(\mathrm{x}_q) \rangle$ held with a *fixed* feature map $\phi_0 : \mathbb{R} \to \mathbb{R}^d$ (independent of the context), then for all contexts the functions $\mathrm{x}_q \mapsto \hat{y}_q(\mathrm{x}_q)$ would lie in the $d$-dimensional linear span of the coordinate functions of $\phi_0$. However, Step 2 shows that by varying the context, SA realizes an infinite set $\mathcal{F}$ of pairwise linearly independent functions in x, which cannot be contained in any finite-dimensional linear subspace. This contradiction rules out the existence of such $(\phi_0, \omega_0)$. □

## C   EXPERIMENTAL DETAILS AND ADDITIONAL RESULTS

In this part of the appendix, we provide detailed descriptions of the experiments in the main text and include additional experimental results.

### C.1   REGRESSION

**Setup details.** We consider two major evaluation scenarios for regression: in-distribution (ID) testing and out-of-distribution (OOD) testing. In the ID scenario, we set $\mathcal{D}_{\mathcal{X}}^{\text{train}} = \mathcal{D}_{\mathcal{X}}^{\text{test}} = \mathcal{D}_{\text{query}}$, and $\mathcal{D}_{\mathcal{F}}^{\text{train}} = \mathcal{D}_{\mathcal{F}}^{\text{test}}$. Specifically, we use the following four classes of functions $\mathcal{F}$:

- Linear functions: $\mathcal{F} = \{ f \mid f(\mathbf{x}) = \mathbf{w}^\top \mathbf{x}, \; \mathbf{w} \in \mathbb{R}^d \}$, where $d = 10$. We sample $\mathbf{x}_1, \ldots, \mathbf{x}_j, \mathbf{x}_q$ and $\mathbf{w}$ independently from the isotropic Gaussian distribution $\mathcal{N}(0, I_d)$, then compute $f(x_i) = \mathbf{w}^\top \mathbf{x}_i$ to construct the prompt. In this setting we use the least squares estimator as the optimal baseline.

- Sparse linear functions: $\mathcal{F} = \{ f \mid f(\mathbf{x}) = \mathbf{w}^\top \mathbf{x}, \; \mathbf{w} \in \mathbb{R}^d, \|\mathbf{w}\|_0 \leq s \}$, where $d = 10$ and $s = 3$. We also sample $\mathbf{x}_1, \ldots, \mathbf{x}_j, \mathbf{x}_q$ and $\mathbf{w}$ independently from $\mathcal{N}(0, I_d)$, and then zero out all but $s$ coordinates of $\mathbf{w}$ uniformly at random. We use the least squares estimator and Lasso, which leverages sparsity with an $\ell_1$-norm regularizer as baselines.

- Two-layer ReLU neural networks: $\mathcal{F} = \{ f \mid f(\mathbf{x}) = \sum_{i=1}^h a_i \sigma(\mathbf{w}_i^\top \mathbf{x}), \; a_i \in \mathbb{R}, \; \mathbf{w}_i \in \mathbb{R}^d \}$, where $\sigma(z) = \max\{0, z\}$ is the ReLU activation function, and $d = 5, \; h = 10$. We sample $\mathbf{x}_i$s and $\mathbf{x}_q$ from $\mathcal{N}(0, I_d)$, along with network parameters $a_i$s from $\mathcal{N}(0, 2/h)$. We sample $\mathbf{w}_i$s from $\mathcal{N}(0, I_d)$, and share them across all tasks in $\mathcal{F}$. The baseline is a two-layer neural network of the same architecture trained on in-context examples using Adam.

- Combination functions: $\mathcal{F} = \{ f \mid f(\mathbf{x}) = \mathbf{w}^\top \Phi(\mathbf{x}), \; \mathbf{w} \in \mathbb{R}^5 \}$, where $\Phi$ is an element-wise combination function. For $\mathbf{x} = [x_1, x_2, x_3, x_4, x_5]$, $\Phi(\mathbf{x}) = [|x_1|, \; x_2^2, \; x_3^3, \; \cos(\pi x_4), \; e^{0.2 x_5}]^\top$. We sample $\mathbf{x}_i$s, $\mathbf{x}_q$ and $\mathbf{w}$ from $\mathcal{N}(0, I_5)$ independently. In this setting, there is no naturally optimal baseline, so we compare only with the Transformer.

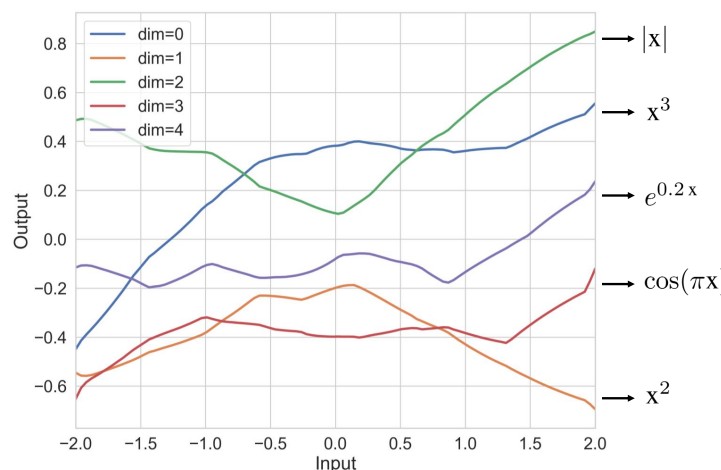

Figure 6: The 5-dimensional sample representation space learned by CoQE for the combination functions.

The latter two classes of nonlinear functions allow the model to reduce ICL difficulty through representation learning, by learning task-invariant $\mathbf{w}_i$s or $\Phi$.

In the OOD scenario, we consider four different cases of distributional shifts under linear functions.

- $\mathcal{D}_{\mathcal{X}}^{\text{train}} \neq \mathcal{D}_{\mathcal{X}}^{\text{test}} = \mathcal{D}_{\text{query}}$. We consider the setting where the prompt inputs $\mathbf{x}_i$s' scale between training and testing is different. We scale them by a factor of $0.8$ or $1.2$.

- $\mathcal{D}_{\mathcal{X}}^{\text{train}} = \mathcal{D}_{\mathcal{X}}^{\text{test}} \neq \mathcal{D}_{\text{query}}$. We sample the context examples from the same distribution as at training time, but sample $\mathbf{x}_q$ from a Gaussian distribution with $3\times$ higher standard deviation.

- $\mathcal{D}_{\mathcal{X}}^{\text{test}} \neq \mathcal{D}_{\mathcal{X}}^{\text{train}} = \mathcal{D}_{\text{query}}$. We fix the sign of each coordinate to be randomly positive or negative for all prompt inputs $\mathbf{x}_i$s, and draw $\mathbf{x}_q$ from $\mathcal{N}(0, I)$ as before.

- $\mathcal{D}_{\mathcal{F}}^{\text{train}} \neq \mathcal{D}_{\mathcal{F}}^{\text{test}}$. We consider scaling the weight vector by a factor of $0.8$ or $1.2$, to capture shifts of task functions.

Through the above diverse evaluation settings, we comprehensively demonstrate that CoQE consistently exhibits stronger ICL capability than a standard Transformer of comparable size on regression tasks.

**Implementation details.** We use Transformer architectures from the GPT-2 family (Radford et al., 2018) as implemented by HuggingFace (Wolf et al., 2020). Specifically, the Transformer baseline we use is configured with an embedding dimension of $64$, $3$ layers, and $2$ attention heads, resulting in a total of $0.2$M parameters. The task encoder of CoQE uses the exact same Transformer configuration. The representation encoder of CoQE consists of a two-layer ReLU network, implemented as a linear projection, followed by a ReLU activation, a LayerNorm, and a second linear layer. For fair comparison, the baseline Transformer's embedding module uses the exact same two-layer ReLU network. During training across the four classes of functions, we use a batch size of $64$ and a learning rate of $5\mathrm{e}-5$. For the three tasks except combination functions, models are trained for $1 \times 10^5$ steps, while the combination task is trained for $2 \times 10^5$ steps due to its increased difficulty. All experiments are conducted on an NVIDIA RTX 4090 GPU.

**Additional results on representation learning.** Our Theorem 3.6 shows that under the dual-space modeling framework, a sufficient set of tasks guarantees a basis-covering sample representation space that the model learns. For empirical validation, we design the task type of two-layer ReLU networks and combination functions, whose different task functions share a common sample representation space in their construction. Figure 3 (a) shows that CoQE indeed achieves a smaller ICL

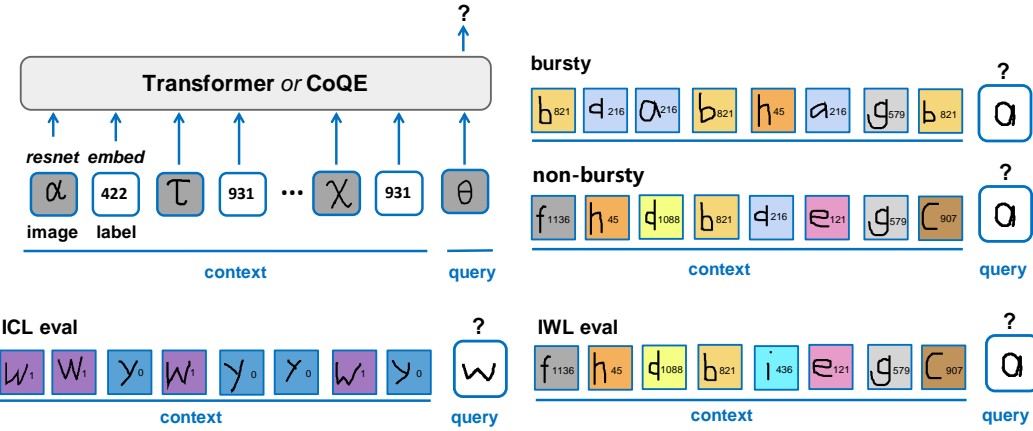

Figure 7: Illustration of the experimental setup for the few-shot classification.

error on these tasks. Furthermore, under the combination functions task, we set the sample representation space dimension of CoQE to 5, matching that of $\Phi$, and directly visualize the 5-dimensional sample representation space learned by CoQE after training (Figure 6). From the figure, we can observe that the five dimensions appear to differentiate in a manner close to the respective transformations of the five dimensions of $\Phi$. Although, due to the equivalence of sample representation spaces under linear transformations, i.e., $f = \mathbf{w}^\top \Phi(\mathbf{x}) = \mathbf{w}^\top H^{-1} \cdot H\Phi(\mathbf{x})$ where $H$ denotes an arbitrary invertible matrix, it is essentially impossible for the model to learn $\Phi$ with perfectly identical scale and shape. The current differentiation can be regarded as another empirical proof of Theorem 3.6 that our dual-modeling could facilitate learning of the basis-covering sample representation space.

## C.2 FEW-SHOT CLASSIFICATION

**Setup details.** To evaluate ICL and IWL abilities under various conditions, we use a synthetic few-shot classification task based on the Omniglot dataset (Lake et al., 2015). The dataset contains $1,623$ character classes, each with 20 samples. Figure 7 provides an illustration of the experimental setup for the few-shot classification, including the overall pipeline, sequences for training, and sequences for testing.

**Implementation details.** In our experiments, we employ ResNets of two sizes (with embedding dimensions $E = 64$ and $E = 512$) to encode images. Both architectures consist of four groups, each containing two residual blocks. The difference lies in the embedding dimensions of each group: for the $E = 64$ ResNet, the four groups produce embeddings of sizes 16, 32, 32, and 64, respectively; for the $E = 512$ ResNet, the sizes are 64, 128, 256, and 512. Although a fully connected layer is appended to the $E = 512$ ResNet to project the final embedding dimension back to 64 before feeding it into the Transformer, it clearly possesses a much stronger capacity for extracting visual sample representations. As a result, the resulting embedding tokens are more expressive. For CoQE, we find that the $E = 64$ ResNet is insufficient for the sample encoder, and therefore adopt the $E = 512$ variant. We also employ two Transformer configurations with different layers: $L = 4$ and $L = 12$. Both variants use an embedding dimension of 64 and 8 attention heads. We have shown that CoQE with only an $L = 4$ Transformer in the task encoder, can match the ICL and IWL performance of an $L = 12$ Transformer. In our experiments, a baseline Transformer with $E = 64$ and $L = 12$ contains approximately 0.9M parameters, while CoQE with $E = 512$ and $L = 4$ has 2.0M parameters.

When training CoQE, we add Gaussian noise to the modified logits to prevent the task encoder's output from collapsing to a static vector. Specifically, the initial noise is sampled from $\mathcal{N}(\mu_0, 1)$, and both the mean and standard deviation are incremented by 1 every $10^4$ training steps. During training of baseline Transformers and CoQE, we use a batch size of 24, a learning rate of $1\mathrm{e} - 4$, and train for $1 \times 10^5$ steps. All experiments are conducted on $8\times$ NVIDIA V100 GPUs.

Table 2: Comparison of model configurations and performance with $P_{\text{bursty}} = 0.9$ and $\alpha = 1$.

| Model | $E$ | $L$ | #Param | ICL | IWL |
|---|---|---|---|---|---|
| Transformer | 64 | 12 | 0.9M | 84.12 | 64.02 |
| CoQE | 256 | 4 | 1.0M | 89.55 | 82.02 |
| CoQE | 512 | 4 | 2.0M | 91.71 | 89.94 |

**Additional results on parameter scale.** As shown above, the only Transformer that can achieve a tradeoff of ICL and IWL has the configurations $E = 64$ and $L = 12$, under the data distribution with $P_{\text{bursty}} = 0.9$ and $\alpha = 1$. CoQE with $E = 512$ and $L = 4$ achieves significantly better performance than the standard Transformer with $E = 64$ and $L = 12$ under all training data distributions, but with a larger number of parameters. To demonstrate the effectiveness of our method under the same parameter scale as the Transformer, we consider CoQE with $E = 256$ and $L = 4$. Specifically, the embedding sizes of its four ResNet groups are $64, 128, 128, 256$, resulting in a total model size of $1.0$M parameters. Under the data distribution of the sweep spot, the results are shown in Table 2. It's obvious that, under the same parameter scale, CoQE still exhibits significantly superior ICL and IWL performance.

**Ablation study.** We present the training curves of CoQE under different levels of noise (Figure 8). It is evident that, in the absence of noise, the model's ICL capability rapidly decays after an initial emergence, accompanied by a similarly rapid increase in IWL performance. Although this observation is not made under a standard Transformer model, we hypothesize that the underlying phenomenon extends beyond model architecture, reflecting the intrinsic properties of the two strategies. ICL is a lightweight, dynamic strategy, whereas IWL is more training-intensive but ultimately more stable. In standard Transformers, where the two strategies are difficult to co-exist, training often leads to a transition from ICL to IWL. In contrast, CoQE enables robust coexistence of both strategies through explicit modeling and learning of the task representation space, as well as the use of Gaussian noise to isolate the task-transformations associated with each strategy.

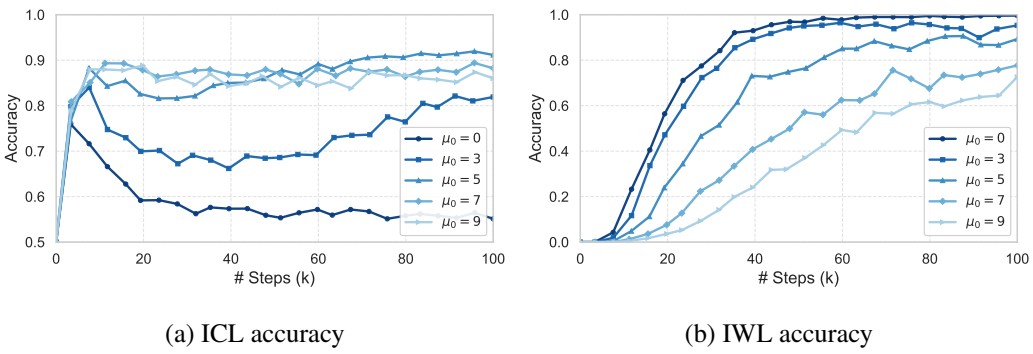

(a) ICL accuracy          (b) IWL accuracy

Figure 8: Learning curves under different noise levels

# D  THE USE OF LARGE LANGUAGE MODELS

In this work, we employed LLMs in a limited capacity to support writing and presentation. Specifically, we used an LLM to help with grammar correction, linguistic polishing, as well as typesetting tables in the appropriate style. All core research contributions were entirely carried out by the authors without LLM involvement.

