# OpenReview forum: "Reconciling In-Context and In-Weight Learning: A Dual-Space Modeling Perspective"
_ICLR.cc/2026/Conference — ICLR 2026 Conference Withdrawn Submission_

### Official Review · Reviewer_ekpH · 2025-10-30

**Soundness:** 3
**Presentation:** 2
**Contribution:** 2
**Rating:** 4
**Confidence:** 4

**Summary:**

This paper aims to reconcile the discrepancy between In-Context Learning (ICL) and In-Weight Learning (IWL) by disentangling the model’s encoding spaces for context and input samples. To achieve this, the authors propose a dual-space modeling framework that explicitly constructs a task representation space as the dual of the sample representation space. They demonstrate that such a structure facilitates ICL through improved representation learning. Building upon this insight, the authors introduce CoQE, a Transformer-based architecture with separate context and query encoders, effectively achieving the desired disentanglement between context and sample representations.

**Strengths:**

1. The paper addresses an important and meaningful research question—the conflict between ICL and IWL.
2. The proposed CoQE architecture introduces a novel design. Beyond empirical validation, the authors also provide a detailed theoretical analysis to support their claims.

**Weaknesses:**

The theoretical presentation lacks clarity and organization:

1. The purpose and implication of **Theorem 3.7** are unclear. It is not evident whether the dual-space formulation achieves a tighter bound than the one presented in this theorem.
2. Although the paper claims to reconcile the conflict between ICL and IWL, **Theorem 3.10** only establishes their entanglement, rather than addressing the nature of their conflict. In fact, such entanglement might allow mutual reinforcement instead of opposition.

Additionally, the computational efficiency of the proposed method is not discussed. The paper omits key details such as the parameter scale and the computational cost associated with the new operations.

Finally, the empirical evaluation only compares CoQE against standard Transformers. It remains uncertain whether the proposed structure outperforms other advanced Transformer variants.

**Questions:**

How to assure that the task representation actually learn the representation of tasks instead of others?

---

> ### Author Response · Authors · 2025-11-13
> **Response to Reviewer ekpH.**
>
> We sincerely thank you for your valuable comments and apologize for any possible lack of clarity in the manuscript. Below we provide clarifications and responses, and we look forward to continued discussion.
>
> W1
> We apologize for any ambiguity in the writing. Theorem 3.7 indeed provides the generalization error bound under the dual-space formulation. Theorem 3.10 aims to formalize the entanglement of Transformers in encoding context and query information. We hypothesize that this entanglement is the underlying cause of the conflict between ICL and IWL, as observed in works such as chan 2025 and singh 2023. By explicitly disentangling this process in CoQE, we effectively resolve the ICL–IWL conflict, which in turn supports the validity of our hypothesis.
>
> W2
> As described in Sec 5.2 (Results), additional experimental results on parameter scale are provided in Appendix C.2 due to space limitations in the main text. The results demonstrate that CoQE consistently exhibits significantly better ICL and IWL performance even under the same parameter scale. We will conduct more extensive experiments in future work to further investigate the influence of model size.
>
> W3
> This is an excellent question. Please allow us some time to complete comparative experiments between CoQE and other Transformer variants.
>
> Q1
> The task representation is the representation obtained by encoding the context portion. In ICL settings, the task itself is entirely defined by the context; therefore, the representation derived from the context can naturally be regarded as the task representation.

---

> > ### Author Response · Authors · 2025-11-18
> > **Additional Experimental Results**
> >
> > We conducted more extensive experiments across parameter scales. The results show that CoQE achieves significantly better ICL and IWL performance at the same parameter budget, regardless of the training data distribution. We also added the Transformer variant Performer[1] as a baseline; since it is not tailored for ICL tasks, it likewise faces a pronounced ICL–IWL trade-off. The experimental results are as follows.
> > ### Transformer
> > | E | L | #Param | P_bursty | α | ICL | IWL |
> > |---|---|---:|---:|---:|---:|---:|
> > | 64  | 12 | 0.9M | 0.9 | 1 | **0.84** | 0.65 |
> > | 64  | 12 | 0.9M | 0.9 | 0 | 0.73 | 0.04 |
> > | 64  | 12 | 0.9M | 0.5 | 0 | 0.65 | 0.18 |
> > | 64  | 12 | 0.9M | 0.1 | 0 | 0.50 | 0.51 |
> > | 64  | 12 | 0.9M | 0.9 | 2 | 0.50 | **0.98** |
> > | 64  | 4  | 0.4M | 0.9 | 0 | 0.50 | 0.15 |
> > | 512 | 12 | 2.4M | 0.9 | 0 | 0.50 | 0.36 |
> > | 512 | 4  | 2.0M | 0.9 | 0 | 0.5 | 0.63 |
> >
> >
> > ### Performer
> > | E | L | #Param | P_bursty | α | ICL | IWL |
> > |---|---|---:|---:|---:|---:|---:|
> > | 64 | 12 | 0.9M | 0.9 | 1 | **0.81** | 0.62 |
> > | 64 | 12 | 0.9M | 0.9 | 0 | 0.72 | 0.02 |
> > | 64 | 12 | 0.9M | 0.5 | 0 | 0.64 | 0.17 |
> > | 64 | 12 | 0.9M | 0.1 | 0 | 0.50 | 0.51 |
> > | 64 | 12 | 0.9M | 0.9 | 2 | 0.50 | **0.92** |
> > | 64 | 4  | 0.4M | 0.9 | 0 | 0.50 | 0.10 |
> >
> >
> > ### CoQE
> > | E | L | #Param | P_bursty | α | ICL | IWL |
> > |---|---|---:|---:|---:|---:|---:|
> > | 256 | 4 | 1.0M | 0.9 | 1 | **0.90**  | 0.82 |
> > | 256 | 4 | 1.0M | 0.9 | 0 | 0.89 | 0.81 |
> > | 256 | 4 | 1.0M | 0.9 | 2 | 0.86 | 0.63 |
> > | 256 | 4 | 1.0M | 0.5 | 0 | 0.88 | 0.82 |
> > | 256 | 4 | 1.0M | 0.1 | 0 | **0.90**  | **0.88** |
> >
> > [1] Rethinking Attention with Performers (Choromanski et al., 2022)

---

### Official Review · Reviewer_BgzZ · 2025-10-30

**Soundness:** 2
**Presentation:** 3
**Contribution:** 2
**Rating:** 2
**Confidence:** 3

**Summary:**

The paper addresses the important challenge of designing autoregressive models that support both in-context learning (ICL) and in-weight learning (IWL). The paper proposes a dual-space modeling framework to allow both ICL and IWL capabilities, providing a theoretical support using linear representation hypothesis. They explicitly model task representation space via the dual space of the sample representation space. To implement this, the paper proposes a new architecture that encodes context and query separately to resolve the representation entanglement, which is identified as the main cause of conflict. The proposed architecture attempts to implement the dual-space theory two spaces interact through inner products. The proposed method is evaluated on a regression task using synthetic data and classification task with Omniglot data. Results show improved results on both ICL and IWL performance.

**Strengths:**

1. The conflict between ICL and IWL in autoregressive transformer models is a relevant topic of research. The idea that samples and tasks can operate in separate but dual representation spaces and the connection with Riesz representation theorem is conceptually novel. The dual-space theoretical framework is well defined covering required formal definitions and assumptions.

2. Results show clear improvements with improved ICL performance without hurting IWL. To support the proposed theory, they further demonstrate empirically that separating context and query stabilizes ICL in both in-distribution and OOD cases. This is shown for both regression and classification tasks.

3. Paper is well-structured and easy to follow.

**Weaknesses:**

1. Authors proposed CoQE architecture to structurally  separate the spaces to resolve the ICL and IWL conflict. However, the dependence on gaussian noise for the classification is very strong to avoid collapse, where performance is close to chance level without the noise injection. The dependence of gaussian noise regularizer parallels with the usage of l2-regularization in [Chan et al. 2022] to balance ICL and IWL, implying that the CoQE benefits step more from regularization than from its theoretical design. This weakens the theoretical claim and advantage of CoQE.

2. Theorem 3.10 equates non-linearity of softmax with the non-existence of factorized dual-space. Non-linearity does not rule out the existence of an equivalent linear form under a suitable feature mapping. The reasoning is limited to a simple setup such as a single softmax-attention layer and doesn’t include a mathematical proof. Empirical studies in the literature such as Han et al. 2025 [a], show that attention-based transformers can exhibit linearly separable task vectors for distinct in-context tasks. This weakens the theorem.
- [a] Emergence and Effectiveness of Task Vectors in In-Context Learning : An Encoder Decoder Perspective, Han et al. 2025.

3. The assumption of a shared linear sample representation across tasks is conceptually good but too strong for real-world multi-task settings. Many real-world tasks in natural language and vision exhibit nonlinear mapping between input and output spaces. This limits the practical applicability of the proposed framework.

Please feel free to clarify if I misunderstood anything mentioned above.

**Questions:**

1. I suggest that the authors demonstrate the advantage of the CoQE architecture with more ICL-IWL tasks which are not dependent on such strong regularizations. This would clarify whether the observed improvements come from the architecture or the regularization.

2. Abstract claims that transformer with softmax self-attention is limiting for the dual-space structure. However, this limitation is not demonstrated clearly in the main text. Authors should clarify what specific theoretical or empirical evidence supports this claim.

3. The paper would benefit from a better connection to previous works in the literature. A comparative discussion with recent studies mentioned below should be included to better position this work. (a) Toward Understanding In-context vs. In-weight Learning (Chan et al. 2025), (b) What Matters for In-Context Learning: A Balancing Act of Look-up and In-Weight Learning (Bratulic et al. 2025), (c) Dual Process Learning: Controlling Use of In-Context vs. In-Weights Strategies with Weight Forgetting (Anand et al. 2025).

4. The paper lacks discussion about certain points such as implications of oversimplified assumptions like shared linear sample space across tasks and limitations of currently used empirical tasks.

Minor remark:
- Figure and table captions should be self-contained clearly conveying the necessary information to understand them.

---

> ### Author Response · Authors · 2025-11-13
> **Response to Reviewer BgzZ.**
>
> We sincerely thank you for your valuable comments and apologize for any possible lack of clarity in the manuscript. Below we provide detailed clarifications and responses, and we look forward to continued discussion.
>
> W1&Q1
> It seems that [1] does not explicitly mention the use of L2 regularization. We assume you are referring to [2], which applies L2 regularization on a ResNet to mitigate the decay of ICL capability. However, it is worth noting that this approach does not achieve a balance between ICL and IWL. As shown in Figure 9 of [2], when L2 regularization is applied, the model’s IWL accuracy remains below 2%. From an empirical perspective, CoQE demonstrates a clear advantage. Furthermore, noise injection is a key part of our dual-space modeling algorithm, designed to prevent the task representation space constructed by the task encoder from collapsing during training. Its effectiveness precisely illustrates the value of maintaining a dynamic and complete task representation space, supporting the theoretical motivation of our model.  Aside from dual-space modeling, we currently have not found a plausible way to apply noise or L2 regularization to any component of a standard Transformer that could be expected to match or surpass the experimental performance of CoQE.
>
> W2&Q2
> Theorem 3.10 identifies the non-existence of a factorized dual-space caused by the nonlinearity of the softmax function. The proof is provided in Appendix B.5, and we apologize for not making this clearer in the main text. Theorem 3.10 is intended to formalize the inherent coupling between context-level and sample-level encoding in Transformers, stemming from the fact that the self-attention (SA) component does not support explicit dual-space modeling. We do not claim that a complete Transformer, given sufficiently complex mappings or feature transformations, can never achieve an equivalent form of dual-space modeling. Proving that a Transformer cannot accomplish a certain representation is evidently much harder than proving it can.  Nonetheless, this does not affect the insight of Theorem 3.10—that Transformers inherently encode context and sample information in a coupled manner—which we identify as a limitation for explicit dual-space modeling and suspect to be the underlying reason for the ICL–IWL conflict.
>
> The work by Han (2025), which you mentioned, is also one of our important references. It shows that in Part-of-Speech tagging and Bitwise Arithmetic tasks, the model learns a linearly separable sample representation space (i.e., the token embeddings after $x_i$) through task encoding, and subsequently performs a mapping from sample representations to answers through task decoding. The function of task decoding in their formulation is analogous to the role of task representation acting on sample representations in ours, which supports the rationale behind our modeling approach.  The key difference, however, is that Han (2025) shows the Transformer performs task encoding and decoding implicitly, whereas we argue that this implicit coupling of task and sample information gives rise to the ICL–IWL conflict. Our goal is to explicitly model the task representation space to resolve this issue.
>
> W3&Q4
> The assumption that multiple tasks share a common linear representation space has been widely used in both theoretical and algorithmic works [3–10]. It is important to emphasize that we do not claim real-world tasks to be linear mappings from input to output spaces; rather, we assume that the input space can be transformed via some nonlinear mapping into a linear representation space.  We acknowledge that the current experimental scenarios are relatively simple, which is indeed a limitation of this work. Extending this modeling approach to broader ICL tasks constitutes an important direction for our future work.
>
> References.
> [1] Data distributional properties drive emergent in-context learning in transformers. (Chan et al., 2022)
> [2] The transient nature of emergent in-context learning in transformers. (Singh et al., 2023)
> [3] Multitask learning. (Caruana et al., 1997)
> [4] Multi-Task Feature Learning. (Argyriou et al., 2006)
> [5] The Benefit of Multitask Representation Learning. (Maurer, 2016)
> [6] Invariant Risk Minimization. (Arjovsky et al., 2019)
> [7] CNN features off-the-shelf. (Razavian et al., 2014)
> [8] Planning-oriented autonomous driving. (Hu et al., 2023)
> [9] Feature contamination: Neural networks learn uncorrelated features and fail to generalize. (Zhang et al., 2024)
> [10] An Information-theoretic Multi-task Representation Learning Framework for Natural Language Understanding. (Hu et al., 2025)

---

> > ### Author Response · Authors · 2025-11-13
> >
> > Q3
> > Appendix A introduces related studies, including theoretical and empirical explorations of ICL and analyses of the relationship between ICL and IWL. For works not mentioned in Appendix A, Bratulic et al. (2025) investigate the effect of the IWL objective on ICL performance and achieve an ICL–IWL trade-off through label noise injection, with accuracies around 50%. Anand et al. (2025) propose the concept of structural ICL and mitigate the conflict between structural ICL and IWL on the head classes of a skewed distribution via temporary forgetting.  In contrast, our approach eliminates the ICL–IWL conflict across all classes under various distributions through explicit modeling and structural modification, achieving consistently superior performance.
> >
> > W5
> > We appreciate your suggestion and will revise the manuscript accordingly in future versions.

---

> > > ### Comment · Reviewer_BgzZ · 2025-11-24
> > >
> > > - Thanks for correcting me concerning the mentioned reference for L2 regularization. I now better understand that noise injection is introduced to maintain non-collapsed/expressive representation space which aligns with the theoretical assumptions. This connection is not clearly articulated in the current paper. A dedicated discussion explaining how noise injection satisfies the theoretical requirement for a dynamic task representation space should be added to the paper.
> > >
> > > - However, I am still uncertain about the generalization of this noising mechanism and the linear representation hypothesis as task becomes more complex and representation dimensionality increases. As also noted by Reviewer cPUL, the current evidence is limited to relatively simpler settings. The noise-injection strategy and the assumed linear structure require more evaluation on more complex tasks and higher-dimensional representation spaces to show their generalization and robustness. I understand that showing evidence on NLP tasks may be difficult, but some intermediate sequence or vision task could be considered as a reasonable extension.

---

> ### Author Response · Authors · 2025-11-29
>
> We thank you for the insightful suggestions and raising score to 4.
> To further verify the generality of our approach and explore its potential extension to NLP tasks,
> we were inspired by Singh and extended our few-shot classification experiments from images
> to token embeddings of large language models.
>
> Specifically, we constructed a fixed token-embedding dataset through the following three steps:
> 1.Subselection: We used the LLaMA3-3B and LLaMA3-8B models,
>     saved their token embedding matrices, and selected all English tokens (33,588 in total).
> 2.Clustering: We applied spherical clustering using FAISS, performing 2,400-way clustering
>     and retaining all clusters with more than 10 members. From these, we randomly selected 1,200 clusters.
> 3.Cluster Sampling: Many clusters contained more than 10  tokens.
>     To maximize intra-class variation, we selected the 10 tokens farthest from each cluster center.
>     Consequently, we obtained 1,200 classes, each containing 10 samples.
>
>
> Compared with Omniglot images, this construction yields classes with larger intra-class variability
> while preserving meaningful semantic relationships relevant to NLP tasks.
> Some example clusters from LLaMA3-3B are shown below:
>
> 1. English | edish  | apanese  | French  | Chinese  | orean  | California  | Russian  | frican  | ustralian
>
> 2. Class |  _class | addClass | _CLASS | className | removeClass | classList | Classifier | hasClass | getClass
>
> 3. ellow | green | Green | iolet | orange | urple | agenta | purple | Pink | greens | _YELLOW
>
>
> We then trained both a Transformer and our CoQE model on this constructed dataset
> following the same setup described in the main text,
> fixing the data distribution parameters at $P_{\text{bursty}} = 0.9$
> and Zipfian exponent $ \alpha = 1$, to approximate natural language distributions.
>
> | **Token Embedding** | **Model**   |  **ICL**  |  **IWL**  |
> | ------------------- | ----------- | :-------: | :-------: |
> | **Llama-3.1-8B**    | Transformer |   50.12   |   96.65   |
> |                     |  **CoQE** | **65.35** | **93.96** |
> | **Llama-3.2-3B**    | Transformer |    49.33   |    98.81   |
> |                     | **CoQE** | **75.01** | **96.57** |
>
>
> The results show that, likely due to the larger intra-class variability,
> the Transformer almost completely fails to acquire ICL ability even after 100k training steps---consistent with Singh’s findings.
> In contrast, CoQE achieves substantially better ICL capability while maintaining strong IWL performance,
> although still lower than in the image-classification experiments.
>
> We believe these results demonstrate that our dual-space modeling of context and sample representations
> can enhance ICL capability and mitigate the ICL--IWL conflict in NLP scenarios where semantic information is richer.
> At the same time, we acknowledge the limitations of the current method in both its theoretical assumptions
> and experimental design; addressing these issues constitutes an important direction for our future work.

---

### Official Review · Reviewer_cPUL · 2025-10-31

**Soundness:** 2
**Presentation:** 2
**Contribution:** 2
**Rating:** 4
**Confidence:** 4

**Summary:**

The paper tackles the observed conflict between in-context learning and in-weight learning . The authors propose that this tension arises because standard Transformers entangle the encoding of context and samples. To address this, they introduce a dual-space modeling framework, where the task representation space is the dual space of the sample representation space. Building on this, they propose CoQE, an architecture that explicitly separates context and query encoding: Predictions are computed via an inner product between task and sample representations.

**Strengths:**

1.	The paper presents a novel theoretical framework to explain the underlying conflict between in-context learning (ICL) and in-weight learning (IWL).
2.	Building on this theory, the authors propose a new architectural design (CoQE) that improves performance and reconciles the trade-off between ICL and IWL.
3.	The figures and visualizations are clear and well-designed.

**Weaknesses:**

1.	The CoQE architecture resembles a dual-tower model, which may compromise the model’s capability for open-ended generation. This limitation reduces its applicability to a broader range of tasks, and additional training would likely be required for different task types.
2.	The evaluation of CoQE is limited to relatively simple regression and classification tasks, which may not be sufficient to demonstrate its effectiveness on more complex or real-world tasks.
3.	The proposed approach relies heavily on a strong linear representation hypothesis, assuming that the task and sample representation spaces form dual linear spaces. The paper lacks sufficient empirical evidence or justification to support this assumption.

**Questions:**

1.	In Section 3.1, the authors state that “each basis often corresponds to an independent attribute or concept.” However, bases in a linear space can be chosen arbitrarily. It is unclear why each basis necessarily corresponds to a specific attribute. Could the authors provide concrete examples to clarify this point?
2.	Could the authors include experiments evaluating CoQE on more complex tasks, such as standard NLP benchmarks, to demonstrate its scalability and general applicability?
3.	Could the authors further explain why a task function can be regarded as a linear function of the sample representation, given that many real-world tasks—especially reasoning tasks requiring chain-of-thought (CoT)—exhibit strong nonlinearity?

---

> ### Author Response · Authors · 2025-11-13
> **Response to Reviewer cPUL.**
>
> We sincerely thank you for your valuable comments and apologize for any possible lack of clarity in the manuscript. Below we provide clarifications and responses, and we look forward to continued discussion.
>
> W1&W2&Q2
> We acknowledge that the experimental scenarios used in this work are relatively simple, similar to many studies in the ICL literature. This is indeed a limitation of our current work, and extending this modeling approach to standard NLP tasks is an important direction for future research. Nevertheless, we believe that CoQE has strong potential for generalization. Natural language generation can also be regarded as a sequential form of classification. In our few-shot classification experiments, CoQE successfully handled over 1,600 categories. With appropriate scaling, it has the potential to extend to natural language generation tasks with vocabularies of tens of thousands of words.
>
> W3
> The assumption of a linear sample representation is not proposed by us; it has already been extensively studied and supported by both empirical and theoretical evidence. Please refer to the section “Linearization in Latent Space” in Appendix A for related work. Our contribution lies in extending this linear representation assumption by introducing the concept of a linear task representation and modeling the dual relationship between the two.
>
> Q1
> Regarding the question about linear representations and corresponding attributes, [1][2] provide illustrative examples. In word embeddings, pairs such as Rep("woman") − Rep("man") and Rep("queen") − Rep("king") correspond to the gender attribute, while pairs such as Rep("China") − Rep("Beijing") and Rep("Russia") − Rep("Moscow") correspond to the capital-city attribute. The entire embedding space can therefore be viewed as being spanned by representation vectors corresponding to various attributes or concepts. In [3], it is further shown that GANs for face generation learn disentangled linear representations in their latent space corresponding to pose, expression, age, and gender. Mathematically, a linear space can have infinitely many bases (via basis transformation), but when considering model representation spaces, selecting basis vectors that carry semantic meaning is clearly more natural.
>
> Q3
> This is an excellent question. As you mentioned, our view is that the task function mapping from inputs to outputs can be regarded as a linear functional defined on the sample representations. Of course, in most cases—whether in simple classification or long reasoning tasks—the task function itself is not linear. Our point is that a well-trained sample encoder can cover the nonlinear parts of these tasks, such that the resulting linear representation space enables the task function to be mapped as a linear operation. In fact, recent studies analyzing model reasoning processes (including chain-of-thought reasoning) have used linear probes from internal representations to correct answers to investigate the underlying mechanisms [4].
>
> References.
> [1] Distributed representations of words and phrases and their compositionality. (Mikolov et al., 2013)
> [2] The Linear Representation Hypothesis and the Geometry of Large Language Models. (Park et al., 2024)
> [3] Interpreting the Latent Space of GANs for Semantic Face Editing. (Shen et al., 2020)
> [4] Physics of language models: Part 2.1, grade-school math and the hidden reasoning process. (Ye et al., 2025)

---

> ### Author Response · Authors · 2025-11-29
>
> To further verify the generality of our approach and explore its potential extension to NLP tasks,
> we were inspired by Singh and extended our few-shot classification experiments from images
> to token embeddings of large language models.
>
> Specifically, we constructed a fixed token-embedding dataset through the following three steps:
> 1.Subselection: We used the LLaMA3-3B and LLaMA3-8B models,
>     saved their token embedding matrices, and selected all English tokens (33,588 in total).
> 2.Clustering: We applied spherical clustering using FAISS, performing 2,400-way clustering
>     and retaining all clusters with more than 10 members. From these, we randomly selected 1,200 clusters.
> 3.Cluster Sampling: Many clusters contained more than 10  tokens.
>     To maximize intra-class variation, we selected the 10 tokens farthest from each cluster center.
>     Consequently, we obtained 1,200 classes, each containing 10 samples.
>
>
> Compared with Omniglot images, this construction yields classes with larger intra-class variability
> while preserving meaningful semantic relationships relevant to NLP tasks.
> Some example clusters from LLaMA3-3B are shown below:
>
> 1. English | edish  | apanese  | French  | Chinese  | orean  | California  | Russian  | frican  | ustralian
>
> 2. Class |  _class | addClass | _CLASS | className | removeClass | classList | Classifier | hasClass | getClass
>
> 3. ellow | green | Green | iolet | orange | urple | agenta | purple | Pink | greens | _YELLOW
>
>
> We then trained both a Transformer and our CoQE model on this constructed dataset
> following the same setup described in the main text,
> fixing the data distribution parameters at $P_{\text{bursty}} = 0.9$
> and Zipfian exponent $ \alpha = 1$, to approximate natural language distributions.
>
> | **Token Embedding** | **Model**   |  **ICL**  |  **IWL**  |
> | ------------------- | ----------- | :-------: | :-------: |
> | **Llama-3.1-8B**    | Transformer |   50.12   |   96.65   |
> |                     |  **CoQE** | **65.35** | **93.96** |
> | **Llama-3.2-3B**    | Transformer |    49.33   |    98.81   |
> |                     | **CoQE** | **75.01** | **96.57** |
>
>
> The results show that, likely due to the larger intra-class variability,
> the Transformer almost completely fails to acquire ICL ability even after 100k training steps---consistent with Singh’s findings.
> In contrast, CoQE achieves substantially better ICL capability while maintaining strong IWL performance,
> although still lower than in the image-classification experiments.
>
> We believe these results demonstrate that our dual-space modeling of context and sample representations
> can enhance ICL capability and mitigate the ICL--IWL conflict in NLP scenarios where semantic information is richer.
> At the same time, we acknowledge the limitations of the current method in both its theoretical assumptions
> and experimental design; addressing these issues constitutes an important direction for our future work.

---

### Official Review · Reviewer_SVff · 2025-10-31

**Soundness:** 2
**Presentation:** 2
**Contribution:** 2
**Rating:** 6
**Confidence:** 3

**Summary:**

This paper studies the interplay between In-Context Learning (ICL) and In-Weight Learning (IWL) in Transformers.
It proposes a dual-space modeling framework that represents these two processes in a task representation space and a sample representation space, linked through the Riesz representation theorem.
The authors argue that the conflict between ICL and IWL originates from the entanglement between context and query encoding in standard self-attention.
To address this issue, they introduce CoQE (Context–Query Encoding Transformer), which explicitly separates context and query encoding pathways.
Experiments on regression and few-shot classification tasks show consistent improvements across both ICL and IWL metrics.

**Strengths:**

- The paper provides a mathematically grounded view of ICL and IWL, introducing a dual-space theoretical framework that distinguishes the task space from the weight (sample) space. This perspective is conceptually novel.
- Theoretical reasoning directly motivates an architectural design (CoQE), forming a coherent pipeline from theory → architecture → experiment.

**Weaknesses:**

- The concept of In-Weight Learning (IWL) should be more explicitly defined within the paper, instead of relying mainly on external references.
- The proposed CoQE structure appears to apply an additional embedding to the context input. From Equation (11), it is unclear how this achieves a true *separation* between context and query encodings, since both still pass through a shared encoder.
- The mapping between ID/OOD performance and IWL/ICL capability is indirect. It is unclear why the authors did not include a no-context baseline (weight-only inference) to explicitly test IWL.

**Questions:**

1. Nonlinear Sample Spaces
   The theoretical assumptions are quite strong and rely on the linearity of the sample space \( $M_F$ \).
   How would the proposed dual-space framework extend to nonlinear or non-convex representation spaces?

2. Memory Interpretation

   I am wondering whether it is appropriate to interpret the interaction between ICL and IWL as analogous to short-term and long-term memory mechanisms.
    Can CoQE be understood as a framework that aims to enhance both types of memory simultaneously, or rather as one that mitigates interference between them?

3. Regression Function Families
   In Section 4.1, the authors write:
   *“Specifically, we use the following four classes of functions F: linear functions, sparse linear functions, two-layer ReLU networks, and combination functions.”*
   Are these referring to target functions used for data generation or to model architectures?
   Figure 3(a) suggests they are target functions, but this should be stated explicitly.

4. Theory–Experiment Gap
   The theoretical part mainly analyzes the entanglement between ICL and IWL and argues that CoQE supports both within the dual-space framework.
   However, the experiments (especially regression and few-shot classification) primarily test ICL generalization, particularly under OOD settings.
   - Are the experiments intended to verify ICL generalization rather than true ICL–IWL coexistence?
   - If claiming “simultaneous improvement,” how is IWL concretely evaluated?
   - Would adding a no-context control (i.e., weight-only inference) better demonstrate CoQE’s preservation of IWL capability?

---

> ### Author Response · Authors · 2025-11-13
> **Response to Reviewer SVff.**
>
> We sincerely thank you for your valuable comments and apologize for any lack of clarity in the manuscript. Below we provide detailed clarifications and responses, and we look forward to continued discussion.
>
> W1.
> In-Weight Learning (IWL) refers to the process during training in which knowledge, strategies, or task rules are consolidated into network parameters through backpropagation and gradient descent. In other words, it represents the standard learning mechanism of neural networks in most scenarios.
>
> W2.
> In our dual-space formulation, the task representation space is defined as the linear functional space of the sample representation space, as specified in Definition 3.2, where $\mathcal{M}_F$ is the domain of $t$. Therefore, the task representation corresponding to a given context must be derived from the sample representation space. Specifically, the context is first encoded into the sample representation space via the sample encoder, and then further mapped into the task representation space via the task encoder.
> The separation between the encoding of context and query is reflected in two aspects:
> (1) the sample encoder adopts a token-wise architecture, meaning that although the context and query share the same encoder, their information does not interact or couple during encoding;
> (2) we explicitly employ the task encoder to map the context into the task representation space, while the query samples are not.
>
> W3 \& Q4.
> The regression experiments follow the classical ICL setups in prior works such as [1], showing that CoQE exhibits stronger in-context learning capability than standard Transformers. Both ID and OOD are scenarios within this experiment, and the regression task does not involve the ICL–IWL conflict.
> The few-shot classification experiments, on the other hand, are designed to demonstrate that CoQE can fully resolve the ICL–IWL conflict phenomenon discussed in [2][3].
> When testing IWL, to avoid unnecessary interference, we adopt the same sequential input format as in ICL evaluation. However, as stated in Sec. 5.2 Setup, the context sequence never contains samples belonging to the same class as the query, which is equivalent to the ``no-context inference'' case you mentioned. We again apologize for the lack of clarity in our original writing.
>
> Q1.
> This is an excellent question. Extending conclusions from linear to nonlinear spaces is inherently difficult, both in mathematics and in AI. Nevertheless, our motivation lies in the strong capability of current models to extract high-quality linear representations from nonlinear data manifolds through complex nonlinear transformations. Please refer to Appendix A, ``Linearization in Latent Space,'' for related discussions.
>
> Q2.
> We fully agree with your interpretation that IWL can be viewed as a form of static long-term memory, while ICL corresponds to a task-dependent working memory. We regard CoQE as a method that alleviates the inherent conflict between these two types of memory during the model’s encoding process, thereby enhancing both simultaneously.
>
> Q3.
> They refer to the target functions used for data generation. As described in Sec. 5.1 Setup, due to space limitations in the main text, detailed definitions of these target functions are provided in Appendix C.1.
>
> [1] What can transformers learn in-context? A case study of simple function classes. (Garg et al., 2022)
> [2] Data distributional properties drive emergent in-context learning in transformers. (Chan et al., 2022)
> [3] The transient nature of emergent in-context learning in transformers. (Singh et al., 2023)

---

### Note · Authors · 2025-12-03

I have read and agree with the venue's withdrawal policy on behalf of myself and my co-authors.